# Towards Generalizable Implicit In-Context Learning with Attention Routing

## Abstract

Implicit in-context learning (ICL) has newly emerged as a promising paradigm that simulates ICL behaviors in the representation space of Large Language Models (LLMs), aiming to attain few-shot performance at zero-shot cost. However, existing approaches largely rely on injecting shift vectors into residual flows, which are typically constructed from labeled demonstrations or task-specific alignment. Such designs fall short of utilizing the structural mechanisms underlying ICL and suffer from limited generalizability. To address this, we propose **In-Context Routing (ICR)**, a novel implicit ICL method that *internalizes* generalizable ICL patterns at the attention logits level. It extracts reusable structural directions that emerge during ICL and employs a learnable input-conditioned router to modulate attention logits accordingly, enabling a train-once-and-reuse framework. We evaluate ICR on 12 real-world datasets spanning diverse domains and multiple LLMs. The results show that ICR consistently outperforms prior implicit ICL methods that require task-specific retrieval or training, while demonstrating robust generalization to out-of-domain tasks where existing methods struggle. These findings position ICR to push the boundary of ICL's practical value.

## 1 Introduction

Large Language Models (LLMs) have been widely adopted for text understanding and generation tasks. As applications broaden, the ability to adapt these models efficiently at inference time has become increasingly important (Brown et al., 2020; Wang et al., 2020b). In-context learning (ICL) is a central mechanism for this adaptation (Dong et al., 2022; Min et al., 2021): by conditioning on a few labeled examples inserted before the query, known as in-context demonstrations (ICDs), the model can perform new tasks without any parameter updates (Wies et al., 2023; Pan, 2023).

Despite its broad adoption, ICL faces two practical limitations: (i) inserting ICDs into the prompt inflates sequence length and inference cost compared to zero-shot use (Peng et al., 2024; Li et al., 2025a), and (ii) performance is brittle, varying with small changes in ICD order or format (Wu et al., 2022; Guo et al., 2024). To address these issues, recent work has explored **implicit ICL**, which converts ICDs into dense vectors that steer intermediate residual flows to approximate the effect of explicit prompting (Hendel et al., 2023; Todd et al., 2023; Liu et al., 2023; Li et al., 2024).

While vector-based implicit ICL offers a new way to simulate ICL behaviors in LLMs, it struggles to generalize across real-world tasks. First, using fixed-size vectors as carriers is inherently restrictive. They can only encode a limited amount of prompt information. Attempts to add new knowledge or transfer it to other models require constructing new vectors. Moreover, this approach lacks a theoretical foundation that is both model-agnostic and input-agnostic. Second, they push LLMs to mimic ICL rather than internalize it, since by the time vectors are applied, the backbone has already settled into a distribution shaped by its own attention dynamics. As a result, they perform well mainly on tasks where explicit ICL already succeeds, but fail to generalize to more challenging cases, such as tasks lacking manually labeled ICDs. To this end, we ask:

*"Can we design an implicit ICL method that enables models to truly **internalize** ICL, thus allowing seamless generalization across diverse ICL scenarios?"*

To examine if there exists a generalizable cross-task ICL pattern, we take explicit multi-task ICL as an empirical probe, which incorporates ICDs from diverse, potentially out-of-domain (OOD) tasks to support those lacking their own labeled examples. This setting provides a unique lens in that it

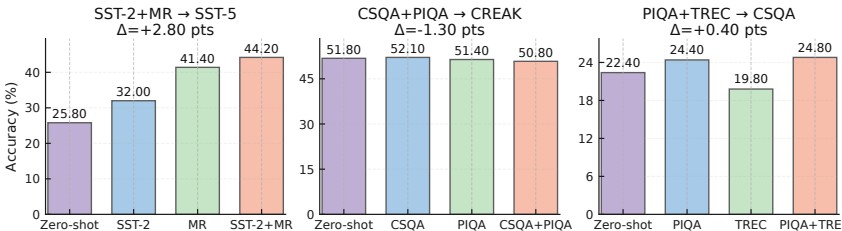

Figure 1: Multi-task ICL on OOD targets. Multi-task few-shot prompting sometimes surpasses both zero-shot and the best single-source few-shot (SST-5, CSQA), but may also degrade performance at times (CREAK). $\Delta$ denotes the difference from the best single-source few-shot prompting.

can sometimes outperform zero-shot prompting and few-shot baselines from single source tasks, but can also yield worse results (Fig 1). This indicates that ICDs from different tasks may embed a latent cross-task pattern beneficial for ICL, yet explicit prompting introduces noise that may obscure it.

Motivated by this, we move deeper than additive residual vectors to investigate the attention space to identify and leverage the cross-domain ICL pattern. We formally analyze how such patterns can be decomposed and embedded directly into attention logits during zero-shot inference, a strategy which we term *attention routing*. Building on this, we propose **In-Context Routing (ICR)**, which extracts the cross-task ICL pattern and employs a router to synthesize it as a low-rank weighted composition, guiding attention computation in a task-adaptive manner.

Empirically, ICR consistently outperforms vector-based implicit ICL baselines across five in-domain and seven out-of-domain (OOD) datasets. It exhibits strong OOD generalization without performance degradation, whereas existing baselines often suffer deficits on certain OOD tasks. ICR also retains key advantages of implicit ICL, including fewer cached parameters and faster inference than few-shot prompting. To the best of our knowledge, ICR is the first implicit ICL method that can be directly adopted for zero-shot inference in diverse new tasks without retrieval or retraining.

Our contributions are three-fold. 1) Recognizing the challenges of post-hoc steering, we propose a new paradigm, *attention routing*. It leverages generalizable ICL patterns that emerge in the attention space across tasks to steer attention logits. 2) Building on this paradigm, we propose **In-Context Routing (ICR)**. Without modifying LLM parameters, ICR introduces a small number of learnable parameters and an end-to-end training strategy that adaptively adjusts routing based on the input query. 3) Extensive experiments validate the effectiveness of ICR, and comprehensive analyses demonstrate that it internalizes ICL patterns while achieving strong adaptivity and generalization.

## 2 ATTENTION ROUTING

This section introduces attention routing, a paradigm that leverages general ICL patterns to intrinsically steer model behavior in zero-shot settings. We begin in Sec. 2.1 by revisiting existing implicit ICL paradigms and their challenges. Sec. 2.2 then presents the formation of attention routing, and Sec. 2.3 analyzes why the general ICL pattern underlying it can be extracted from LLM attention.

### 2.1 PRELIMINARIES AND CHALLENGES OF EXISTING WORK

An ICL prompt input $\mathbf{p}$ to the LLM is typically constructed from several labeled examples serving as in-context demonstrations (ICDs) and a query sample. We denote it as $\mathbf{p} = [\mathbf{D}, x_q]$, where $\mathbf{D} = \{(x_i, y_i)\}_{i=1}^n$ represents the set of $n$ ICDs and $x_q$ is the query sample. The model is expected to infer the input-label mappings illustrated by the ICDs and then predict the label associated with the query sample. Extensive studies have shown that the multi-head attention (MHA) module in transformer-based models plays a central role in learning from $\mathbf{D}$ (Olsson et al., 2022; Chen et al., 2024), which performs a soft query-conditioned retrieval over the ICDs to acquire key knowledge.

**Vector-based implicit ICL** replaces explicit token-level ICDs with dense vectors injected into the model's internal layers. They find that ICDs can be viewed as additive modifications to the MHA outputs in the zero-shot setting and steer the model using vectors that represent ICL (Peng et al., 2024). A typical approach is to add the activation differences induced by ICDs as shift vectors to the zero-shot hidden states. Formally, given an LLM with hidden dimension $d$ and an input sequence of $T$ tokens, the MHA output $\tilde{\mathbf{h}}_t^l$ of token $t$ at layer $l$ is given by:

$$\mathbf{h}^l = \text{Concat}_h\big(\text{softmax}(\mathbf{A}^{l,h})V^{l,h}\big) = \text{Concat}_h\big(\text{softmax}(\frac{Q^{l,h}K^{l,h\top}}{\sqrt{d_k}} + \mathbf{M})V^{l,h}\big), \quad (1)$$

$$\tilde{\mathbf{h}}_t^l = \mathbf{h}_t^l + \beta^l \cdot \mathbf{V}_{\text{shift}}^l, \tag{2}$$

where $\mathbf{h}^l \in \mathbb{R}^{T \times d}$ denotes the zero-shot MHA output at layer $l$ and $Q^{l,h}, K^{l,h}, V^{l,h} \in \mathbb{R}^{T \times d_k}$ are head projections of the final output from layer $l - 1$. $d_k$ is the dimensionality of each head and $\mathbf{M}$ is a causal mask. $\mathbf{A}^{l,h} \in \mathbb{R}^{T \times T}$ is the matrix of attention logits at layer $l$ and head $h$. $\mathbf{V}_{\text{shift}}^l \in \mathbb{R}^d$ is a shift vector. It is typically derived from explicit ICL, for example, by averaging the hidden states of $n$ ICDs' last tokens. The scalar coefficient $\beta^l \in \mathbb{R}$ controls the magnitude of this shift.

**Challenges.** The steering approach in Eq. 2, while effective for task-specific adaptation, is **inherently limited in generalizability**. It operates in a post-hoc manner where a shift vector is directly injected into the residual stream. Such additive interventions cannot structurally control how information flows, and thus often remain tied to task-specific representations. In contrast, more generalizable ICL patterns are expected to lie in how queries are routed through alternative attention paths. This motivates our hypothesis that modulating the matching geometry in the attention space, rather than perturbing outputs post hoc, better reflects the mechanism of ICL, where query tokens attend to the most relevant directions (Olsson et al., 2022; Cho et al., 2025). We therefore argue that attention logits provide a principled basis for extracting task-agnostic and transferable ICL patterns. Since it intrinsically directs model attention to desired routes, we refer to steering attention logits during zero-shot inference as *attention routing*.

### 2.2 How Attention Routing Works

As shown in Eq. 1, attention logits are governed by query-key interactions, making their projections a natural entry point for mining ICL patterns. Specifically, we treat the last token of each ICL prompt as the integration point where contextual information is consolidated. By examining its query and key projections, we can capture systematic shifts induced by the presence of ICDs across diverse tasks. These shifts give rise to a low-dimensional subspace capturing generalizable ICL dynamics. To recover this subspace, we first perform explicit ICL across multiple domains to obtain high-dimensional mixed-domain

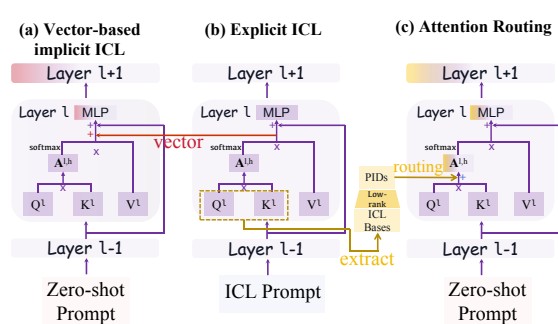

Figure 2: Illustration of attention routing compared with vector-based implicit ICL, with head-level details omitted for clarity.

attention representations. Specifically, we iteratively input ICL prompts into the LLM, each prompt containing ICDs and a query sample from the same domain. We then collect the last-token $Q$ and $K$ projections across domains and stack them to form two **ICL bases**. Principal Component Analysis (PCA) is applied separately to each base, yielding two sets of layer-wise **Principal ICL Directions (PIDs)**, denoted for each layer $l$ as $U_q^l, U_k^l \in \mathbb{R}^{d \times r}$, where $r$ is the rank of the PID subspace.

We define a *routing vector* $\alpha^l \in \mathbb{R}^r$ that assigns weights to the PIDs at layer $l$. $\alpha^l$ controls the strength with which each PID modulates the attention. During zero-shot inference, the layer-level query and key projections are formed by concatenating the per-head projections $Q_{\text{zs}}^{l,h}, K_{\text{zs}}^{l,h} \in \mathbb{R}^{T \times d_k}$, yielding $Q_{\text{zs}}^l, K_{\text{zs}}^l \in \mathbb{R}^{T \times d}$. The routing vector specifies a low-rank modulation of the attention logits and thereby biases the attention dynamics toward the extracted PIDs:

$$\Delta \mathbf{A}^l = \left( Q_{\text{zs}}^l U_q^l \right) \text{diag}(\alpha^l) \left( K_{\text{zs}}^l U_k^l \right)^\top \in R^{T \times T}. \tag{3}$$

The layer-level bias $\Delta \mathbf{A}^l$ is shared across all $H$ heads in layer $l$, so that each head's routed logits become $\tilde{\mathbf{A}}^{l,h} = \mathbf{A}^{l,h} + \Delta \mathbf{A}^l$. Figure 2 shows the key difference between attention routing and vector-based implicit ICL. We further provide a kernel-based perspective in Appendix A.1.

### 2.3 Why PIDs Capture General ICL Pattern

We now explain why the low-dimensional subspaces defined by the PID sets $\{U_q^l\}_{l=1}^L$ and $\{U_k^l\}_{l=1}^L$, derived from multi-domain ICL, capture a general attention pattern to enable ICL. As described in Sec. 2.2, at each layer we derive two ICL bases, $Q$ and $K$, by stacking projections across multiple domains. Considering the rows of $Q$ from a particular domain d, we can model its covariance under the Spiked Covariance Model (Johnstone, 2001) (see Appendix A.2) as a mixed spiked form:

$$\Sigma_Q^{(\mathrm{d})} = S_q \Lambda_q S_q^\top + B_{q,\mathrm{d}} \Gamma_{q,\mathrm{d}} B_{q,\mathrm{d}}^\top + \sigma^2 I, \tag{4}$$

where $S_q \in \mathbb{R}^{d \times r}$ captures a low-dimensional subspace of attention structures shared across domains, while $B_{q,\mathrm{d}}$ encodes domain-specific variations with energy $\Gamma_{q,\mathrm{d}}$. $\sigma^2 I$ represents isotropic noise. An analogous decomposition holds for $K$. Let $\{\mathcal{D}_1, \ldots, \mathcal{D}_{\mathrm{D}}\}$ denote all $\mathrm{D}$ domains involved in the ICDs. We define the pooled covariance of $Q$ as:

$$\widehat{\Sigma}_Q \;=\; \frac{1}{N} \sum_{\mathrm{d}=1}^{\mathrm{D}} \sum_{i \in \mathcal{D}_{\mathrm{d}}} Q_i Q_i^\top, \qquad N = \sum_{\mathrm{d}=1}^{\mathrm{D}} |\mathcal{D}_{\mathrm{d}}|. \tag{5}$$

We compute the expectation of $\widehat{\Sigma}_Q$ and expand it under the mixed spiked form defined in Eq. 4 as:

$$\mathbb{E}[\widehat{\Sigma}_Q] = S_q \Lambda_q S_q^\top + \sigma^2 I + \frac{1}{N} \sum_{\mathrm{d}=1}^{\mathrm{D}} |\mathcal{D}_{\mathrm{d}}| \, B_{q,\mathrm{d}} \Gamma_{q,\mathrm{d}} B_{q,\mathrm{d}}^\top. \tag{6}$$

The same expansion holds for $\widehat{\Sigma}_K$. The first term corresponds to the ICL structure shared across domains, while the last term aggregates domain-specific variations. If the domain-specific subspace set $\{B_{q,\mathrm{d}}\}$ are sufficiently diverse and lack consistent alignment, their aggregate contribution averages out toward isotropy. In this case, they primarily increase background variance rather than forming dominant eigen-directions. In contrast, the shared component $S_q \Lambda_q S_q^\top$ accumulates consistently across all domains. In this way, PIDs obtained by PCA on multi-domain ICL bases recover a domain-stable ICL pattern. Appendix A.3 provides perturbation analysis supporting this claim, and Appendix A.4 further examines the validity of the extracted pattern in OOD settings.

## 3 METHOD

Building on the foundation of attention routing, we propose a new implicit ICL method, termed **In-Context Routing (ICR)**. ICR leverages attention routing to dynamically integrate extracted Principal ICL Directions (PIDs) into the attention space, thereby enhancing zero-shot inference of LLMs. We instantiate ICR in three stages: (i) PIDs extraction across multiple domains, (ii) a query-conditioned router that determines low-rank routing vectors and head gates, and (iii) multi-objective training that combines supervision with stable and sparse routing. The pipeline of ICR is illustrated in Figure 3 and presented in pseudocode in Appendix C.

### 3.1 PRINCIPAL ICL DIRECTIONS EXTRACTION

To implement ICR, we first extract the ICL bases from the model's ICL across multiple domains, along with the PIDs contained within them. For $\mathrm{D}$ domains, we construct a set of ICL prompts for each domain d, denoted as $\mathcal{P}_{\mathrm{d}}$. Let $N = \sum_{\mathrm{d}=1}^{\mathrm{D}} |\mathcal{P}_{\mathrm{d}}|$ be the total number of constructed ICL prompts across all domains. These prompts are fed into the LLM domain by domain. During inference of the $i$-th prompt from domain d, we extract the query and key projections of its *last token* in the layer $l$ and the head $h$, denoted $q_{\mathrm{d},i}^{l,h}, k_{\mathrm{d},i}^{l,h} \in \mathbb{R}^{1 \times d_k}$. We then concatenate them across heads to obtain layer-level vectors $q_{\mathrm{d},i}^l = \mathrm{Concat}_h \, q_{\mathrm{d},i}^{l,h} \in \mathbb{R}^{1 \times d}$ and $k_{\mathrm{d},i}^l = \mathrm{Concat}_h \, k_{\mathrm{d},i}^{l,h} \in \mathbb{R}^{1 \times d}$. Finally, these vectors are stacked across prompts and domains to yield the ICL bases across $\mathrm{D}$ domains.

$$\widetilde{Q}^l = \mathrm{stack}_{\mathrm{d}=1}^{\mathrm{D}} \mathrm{stack}_{i=1}^{|\mathcal{P}_{\mathrm{d}}|} q_{\mathrm{d},i}^l \in \mathbb{R}^{N \times d}, \qquad \widetilde{K}^l = \mathrm{stack}_{\mathrm{d}=1}^{\mathrm{D}} \mathrm{stack}_{i=1}^{|\mathcal{P}_{\mathrm{d}}|} k_{\mathrm{d},i}^l \in \mathbb{R}^{N \times d}. \tag{7}$$

From $\widetilde{Q}^l, \widetilde{K}^l$ constructed above, we then obtain the top-$r$ principal directions by PCA to form the PIDs $U_q^l, U_k^l \in \mathbb{R}^{d \times r}$. These PIDs serve as reusable routing directions for downstream control of attention logits during both training and inference.

### 3.2 QUERY-CONDITIONED ROUTER

After obtaining the PIDs, our goal is to construct the attention routing form introduced in Sec. 2.2. To apply these cross-domain ICL patterns during inference on various input queries, we employ a learnable router to optimize the routing process. Given a query sample $x$, it is fed into the LLM and a frozen text encoder, which produces a representation $\mathrm{E}(x)$. $\mathrm{E}(x)$ is then passed to a two-branch router consisting of two two-layer MLPs, $g_{\theta_\alpha}$ and $g_{\theta_\gamma}$. The two branches generate a routing matrix $\alpha(x) \in \mathbb{R}^{L \times r}$ and a gating matrix $\gamma(x) \in \mathbb{R}^{L \times H}$ in parallel, computed as

$$\alpha(x) = \tanh\big(g_{\theta_\alpha}(\mathrm{E}(x))\big) \in \mathbb{R}^{L \times r}, \tag{8}$$

$$\gamma(x) = \sigma\big(g_{\theta_\gamma}(\mathrm{E}(x))\big) \in \mathbb{R}^{L \times H}, \tag{9}$$

Figure 3: Pipeline of In-Context Routing (ICR). (a) We perform ICL across multiple domains to extract PIDs, which can be stored and reused. (b) We train the router with zero-shot inputs while keeping the LLM frozen, and it generates query-conditioned matrices to control the routing.

where $\sigma(\cdot)$ denotes the sigmoid function. $\alpha^l(x) \in \mathbb{R}^{1 \times r}$ denotes the $r$-dimensional routing vector at layer $l$, and $\gamma^{l,h}(x) \in \mathbb{R}^{1 \times 1}$ provides head-specific gates at layer $l$ and head $h$. Together, $\alpha(x)$ adaptively amplifies or attenuates the extracted PIDs according to query semantics, and $\gamma(x)$ regulates the contributions of individual heads. They jointly produce a low-rank bias that leverages the PIDs in a query-conditioned manner to modulate the zero-shot attention logits for input $x$:

$$\tilde{\mathbf{A}}^{l,h}(x) = \mathbf{A}^{l,h}(x) + \gamma^{l,h}(x) \left( Q_{\mathbf{zs}}^l U_q^l \right) \operatorname{diag}\left( \alpha^l(x) \right) \left( K_{\mathbf{zs}}^l U_k^l \right)^\top, \tag{10}$$

Again, $Q_{\mathbf{zs}}^l, K_{\mathbf{zs}}^l \in \mathbb{R}^{T \times d}$ are the concatenation of head-level projections $Q_{\mathbf{zs}}^{l,h}, K_{\mathbf{zs}}^{l,h} \in \mathbb{R}^{T \times d_k}$. $\tilde{\mathbf{A}}^{l,h}(x)$ is then applied to the subsequent attention computation and final answer generation.

### 3.3 TRAINING OBJECTIVE

During ICR training, only the router parameters $(\theta_\alpha, \theta_\gamma)$ are updated. The training set is constructed by sampling and mixing subsets from each domain $\mathcal{D}_\mathsf{d} \in \mathcal{D}$. We then construct mini-batches of size $B$, each denoted as $\{(x_i, y_i), \mathcal{D}_\mathsf{d}\}_{i=1}^B$, where $(x_i, y_i)$ is an input–label pair and $\mathcal{D}_\mathsf{d}$ indicates its domain. Within each mini-batch, we obtain (i) the zero-shot output $p_i^{\mathsf{zs}} \in \mathbb{R}^{|\mathcal{V}|}$ and (ii) the output under ICR $p_i^{\mathsf{ICR}} \in \mathbb{R}^{|\mathcal{V}|}$ of the generated answer, where $\mathcal{V}$ is the model's vocabulary.

**(1) Supervised cross-entropy.** To provide solid semantic supervision for training ICR, we first adopt the standard cross-entropy loss. For each input and its ground-truth label $(x_i, y_i)$, the loss is:

$$\mathcal{L}_{\mathrm{CE}} = -\frac{1}{B} \sum_{i=1}^B \log P^{\mathsf{ICR}}(y_i | x_i). \tag{11}$$

**(2) Confidence alignment.** We encourage routed predictions to be at least as confident as zero-shot ones via an entropy drop objective. This prevents the router from taking a shortcut of producing over-uncertain predictions and ensures routed inference does not reduce confidence:

$$\mathcal{L}_{\mathrm{conf}} = \frac{1}{B} \sum_{i=1}^B \operatorname{ReLU}\Big( H\big(\operatorname{softmax}(p_i^{\mathsf{ICR}})\big) - H\big(\operatorname{softmax}(p_i^{\mathsf{zs}})\big) \Big), \quad H(q) = -\sum_{v \in \mathcal{V}} q_v \log q_v. \tag{12}$$

**(3) Sparse routing.** We regularize the per-layer routing vectors $\alpha^l(x) \in \mathbb{R}^r$ and gates $\gamma^l(x) \in \mathbb{R}^H$ to encourage sparsity in the modulation that ICR introduces to MHA. Because later layers are closer to the final prediction and should depend on fewer but more decisive routing directions, we scale the sparsity penalty with a layer-dependent weight $w^l$ that increases linearly with depth:

$$\mathcal{L}_{\mathrm{spar}} = \mathbb{E}_x \left[ \frac{1}{L} \sum_{l=1}^L w^l \frac{\|\alpha^l(x)\|_1}{r} \right], \qquad \mathcal{L}_{\mathrm{gate}} = \mathbb{E}_x \left[ \frac{1}{L} \sum_{l=1}^L \frac{\|\gamma^l(x)\|_1}{H} \right]. \tag{13}$$

The final training objective is a weighted combination of the above three terms:

$$\mathcal{L} = \mathcal{L}_{\mathrm{CE}} + \lambda_{\mathrm{conf}} \mathcal{L}_{\mathrm{conf}} + \lambda_{\mathrm{spar}} \mathcal{L}_{\mathrm{spar}} + \lambda_{\mathrm{gate}} \mathcal{L}_{\mathrm{gate}}, \tag{14}$$

where $\lambda_{\mathrm{conf}}$, $\lambda_{\mathrm{spar}}$, and $\lambda_{\mathrm{gate}}$ are hyperparameters that weight each corresponding loss term.

### 3.4 INFERENCE

During inference, ICR is implemented by adding low-rank biases to the attention logits of the corresponding heads, as defined in Eq. 10, while keeping the backbone parameters frozen. When given a zero-shot prompt, ICR adaptively forms $\tilde{\mathbf{A}}^{l,h}(x)$, which the model then uses for subsequent prefilling and complete decoding. The entire procedure operates purely on the query representations and does not access any label space or task-specific supervision at test time. In this way, ICR implicitly equips zero-shot inference with the effect of ICL by fundamentally routing attention dynamics along shared structural directions via query-conditioned composition, regardless of whether the input belongs to a domain seen during training.

## 4 EXPERIMENTS

### 4.1 SETUPS

This section introduces the models employed and the settings for cross-domain collections, training, and evaluation of ICR. Further details are provided in Appendix D.

**Models**  ICR is evaluated on three open-source LLMs: Llama2-7B (Touvron et al., 2023), Qwen2.5-7B (Yang et al., 2025), and Llama3.1-8B (Grattafiori et al., 2024). All ablation and analysis studies are conducted on Llama2-7B as an example.

**Cross-domain collections**  We consider five datasets with distinct task types: AGNews (Zhang et al., 2015), SST-2 (Socher et al., 2013), TREC (Li & Roth, 2002), CSQA (Talmor et al., 2019), and PIQA (Bisk et al., 2020), and treat each dataset as a separate domain. For each dataset, we construct ICL prompts by first sampling a query and a balanced set of ICDs, both from the training split, where 5 ICDs are drawn from each class of the same dataset. We construct 10k prompts for AGNews and 5k prompts for each of the remaining datasets. After feeding each prompt into the LLM, we extract the layer-wise $Q$ and $K$ representations of the last token. They are aggregated across all prompts to obtain per-layer ICL bases as in Eq. 7, enabling PIDs extraction via PCA. More details about ICL prompts construction during collection are presented in Appendix D.

**Training**  We train the router on a set of 25k queries, obtained by randomly sampling 5k queries from the training split of each of the five datasets and shuffling them together. Each query is first encoded by a frozen MiniLM encoder (Wang et al., 2020a), and its pooled representation is fed into the router. The ICR is applied only to the **last** one-third of the LLM layers. We set $\lambda_{\text{conf}} = 0.01$, $\lambda_{\text{spar}} = 10^{-3}$, and $\lambda_{\text{gate}} = 0.02$ during training.

**Evaluation**  We evaluate on 500 randomly sampled test instances (or the full set if smaller) using dataset-specific prompts and a batch size of 4. Each experiment is run with three seeds, and we report the average results. We treat the five datasets used for training as **in-domain (ID)** and select seven additional datasets for out-of-domain (OOD) evaluation. Based on their task similarity to the training datasets, we further categorize them into **near OOD** and **far OOD**. The near OOD datasets include SST-5 (Socher et al., 2013), MR (Pang & Lee, 2005), and MRPC (Dolan & Brockett, 2005), while the far OOD datasets include CB (De Marneffe et al., 2019), COPA (Roemmele et al., 2011), CREAK (Onoe et al., 2021), and AI2SciE (Clark et al., 2018). In addition to zero-shot and few-shot prompting, we choose three vector-based methods with calibration or training as baselines: I2CL (Li et al., 2024), LIVE (Peng et al., 2024), and M$^2$IV (Li et al., 2025a). We further compare the in-domain performance of ICR with five training-free methods: TV (Hendel et al., 2023), FV (Todd et al., 2023), ICV (Liu et al., 2023), ELICIT (Wang et al., 2024a), and IV (Liu & Deng, 2025).

### 4.2 MAIN RESULTS

As shown in Table 1, ICR closely matches and can even surpass few-shot prompting on ID tasks. It consistently outperforms all implicit ICL baselines. Notably, these methods often require additional task-specific retrieval or training, whereas ICR operates in a train-once-and-reuse manner, further highlighting its practical value. On OOD tasks, multi-task few-shot prompting is unstable, performing well on some tasks but collapsing on others, which corroborates the limitations observed in Figure 1. By design, vector-based implicit ICL inherits the drawbacks of explicit ICL, leading to higher failure rates. In contrast, ICR improves over the best implicit baseline by +3.0% on Llama2-7B and +6.5% on Qwen2.5-7B, and even surpasses few-shot prompting by +2.7% on Qwen2.5-7B. These results establish ICR as a generalizable paradigm for implicit ICL. We also compare ICR with vector-based ICL variants that inject dataset-specific vectors into hidden states

Table 1: Baseline comparison across benchmarks. [*]For ID datasets, few-shot uses 5-shot balanced sampling per class. For OOD datasets, we adopt multi-task few-shot prompting where each ID dataset provides 3-shot ICDs. The *Collapse* column reports the number of cases where a method underperforms the zero-shot baseline. Results on Llama3.1-8B are shown in Appendix E.1.

| Method | In-Domain (ID) | | | | | Near OOD | | | Far OOD | | | | Overall | |
|---|---|---|---|---|---|---|---|---|---|---|---|---|---|---|
| | AG | SST-2 | TREC | CSQA | PIQA | SST-5 | MR | MRPC | CB | COPA | CREAK | AI2SciE | Average | Collapse |
| | | | | | | | **Llama2-7B** | | | | | | | |
| Zero-shot | 67.0 | 78.6 | 56.6 | 22.4 | 52.2 | 25.8 | 72.2 | 44.4 | 37.5 | 63.0 | 51.8 | 34.8 | 50.5 | – |
| Few-shot[*] | 81.0 | 95.2 | 84.6 | 58.0 | 59.8 | 37.4 | 98.6 | 68.2 | 41.1 | 82.0 | 50.8 | 45.4 | 66.8 | 1 |
| I2CL | 85.5 | 86.0 | 78.6 | 23.8 | 55.6 | 27.6 | 71.6 | 42.4 | 38.2 | 63.6 | 52.6 | 35.0 | 55.0 | 2 |
| LIVE | 86.0 | 86.2 | 81.0 | 24.2 | 56.4 | 32.8 | 73.8 | 47.6 | 40.8 | 64.8 | 51.0 | 34.6 | 56.6 | 2 |
| M²IV | 86.4 | **86.4** | 81.5 | **24.8** | 56.8 | 30.8 | 74.0 | 46.0 | 42.6 | 64.8 | 54.0 | 35.2 | 56.9 | **0** |
| **ICR** | **86.6** | **86.4** | 83.8 | 24.8 | 57.0 | 38.6 | 79.8 | 53.4 | 46.4 | 68.0 | 56.4 | 37.2 | 59.9 | **0** |
| | | | | | | | **Qwen2.5-7B** | | | | | | | |
| Zero-shot | 66.8 | 54.0 | 65.8 | 80.4 | 76.2 | 31.4 | 64.4 | 72.4 | 83.9 | 92.0 | 77.8 | 90.4 | 71.3 | – |
| Few-shot[*] | 80.2 | 95.6 | 67.6 | 82.2 | 86.0 | 37.2 | 70.2 | 76.2 | 83.9 | 95.0 | 59.7 | 95.8 | 77.5 | 1 |
| I2CL | 77.0 | 86.4 | 68.6 | 81.6 | 81.2 | 34.6 | 69.0 | 70.8 | 80.6 | 92.6 | 74.8 | 91.8 | 75.6 | 3 |
| LIVE | 79.0 | 87.8 | 70.4 | 81.6 | 82.0 | 30.8 | 68.6 | 69.4 | 81.0 | 93.2 | 72.8 | 91.8 | 75.7 | 4 |
| M²IV | 79.6 | 89.0 | **70.8** | 81.8 | 82.5 | 31.6 | 71.2 | 71.0 | 76.0 | 93.5 | 74.6 | 92.4 | 76.2 | 3 |
| **ICR** | **80.4** | **91.0** | 70.6 | **82.0** | **82.6** | 41.4 | 89.4 | 73.2 | 84.6 | 95.0 | 79.2 | 93.2 | 80.2 | **0** |

Table 2: Baseline comparison on in-domain benchmarks.

| Method | Llama2-7B | | | | | | Qwen2.5-7B | | | | | |
|---|---|---|---|---|---|---|---|---|---|---|---|---|
| | AG | SST-2 | TREC | CSQA | PIQA | Overall | AG | SST-2 | TREC | CSQA | PIQA | Overall |
| TV | 82.8 | 83.4 | 73.4 | 22.6 | 53.0 | 63.0 | 70.4 | 78.2 | 64.6 | 80.6 | 74.6 | 73.7 |
| FV | 83.6 | 82.8 | 72.8 | 22.4 | 52.5 | 62.8 | 68.4 | 76.8 | 66.2 | 78.8 | 80.0 | 74.0 |
| ICV | 83.6 | 84.2 | 74.2 | 23.0 | 52.8 | 63.5 | 74.6 | 83.0 | 67.2 | 81.3 | 77.2 | 76.7 |
| ELICIT | 84.0 | 84.4 | 75.8 | 22.4 | 53.9 | 64.1 | 70.4 | 78.5 | 65.0 | 79.2 | 76.4 | 74.3 |
| IV | 83.8 | 85.6 | 73.8 | 23.2 | 54.6 | 64.2 | 73.8 | 78.4 | 66.0 | 81.2 | 77.8 | 75.4 |
| **ICR** | **86.6** | **86.4** | **82.2** | **24.8** | **57.0** | **67.4** | **80.4** | **91.0** | **70.6** | **82.0** | **82.6** | **81.2** |

(Table 2). These ad-hoc methods lack transferability and are evaluated only on five ID datasets. ICR consistently outperforms them by a clear margin, indicating that attention routing captures deeper and more general ICL patterns. Appendix E.2 further compares ICR with few-shot LoRA (Hu et al., 2021), a PEFT-based finetuning method. Appendix F provides an efficiency analysis of ICR.

## 4.3 ABLATION STUDY

In this section, we provide ablations on the extraction of PIDs and the key components of ICR. Further analyses on the strategy for sampling ICDs in constructing the ICL bases and on the effect of routing layer positions are presented in Appendix G.2 and Appendix G.3.

**PIDs Extraction** To understand the role of PIDs extraction, we conduct two ablations (Table 3). First, we vary the PCA rank $r \in 4, 8, 12$. Compared to $r = 8$, reducing $r$ to 4 improves in-domain and near-OOD results but sharply reduces far-OOD accuracy, as the stronger bottleneck regularizes domain signals but suppresses the diversity needed for transfer. Increasing $r$ to 12 consistently hurts, likely due to the enlarged subspace introducing degrees of freedom that re-

Table 3: Ablation on PIDs Extraction. "R.O." denotes the replacement of PIDs with a random orthogonal basis. Scores are averaged within ID, near-OOD, and far-OOD groups.

| Setting | ID | Near OOD | Far OOD |
|---|---|---|---|
| r=4 (PCA) | **67.8** | **57.5** | 45.6 |
| r=8 (PCA) | 67.7 | 57.3 | **52.0** |
| r=12 (PCA) | 53.2 | 54.4 | 43.4 |
| r=8 (R. O.) | 63.9 | 48.1 | 46.7 |

main under-trained. Second, we replace PCA with a random orthogonal basis ($r = 8$). While ID performance remains close to PCA, both near- and far-OOD collapse. This shows that low-rank routing alone is insufficient: OOD robustness crucially depends on aligning with meaningful ICL directions extracted by PCA. A more detailed study on PIDs extraction is provided in Appendix G.1.

**Key Components** Table 4 presents ablations of the key components of ICR, including the auxiliary loss terms in Eq. 14 and the query-conditioned modulation of $\alpha$ and $\gamma$. Dropping $\mathcal{L}_{\text{spar}}$ or $\mathcal{L}_{\text{gate}}$ has little impact on ID and near-OOD tasks but leads to clear degradation on far-OOD datasets, consistent with their role in constraining over-intervention and enhancing transferability. Removing the confidence-alignment loss $\mathcal{L}_{\text{conf}}$ produces less systematic changes, suggesting that its primary effect is stabilizing routing by suppressing entropy inflation rather than directly improving ICL accuracy. For $\alpha$ and $\gamma$, we preserve their magnitude but redistribute it uniformly across PID directions or

Table 4: Ablation of key components in ICR.

| Ablation | In-Domain (ID) | | | | | Near OOD | | | Far OOD | | | |
|---|---|---|---|---|---|---|---|---|---|---|---|---|
| | AG | SST-2 | TREC | CSQA | PIQA | SST-5 | MR | MRPC | CB | COPA | CREAK | AI2SciE |
| FULL | **86.6** | 86.4 | 83.8 | 24.8 | **57.0** | **38.6** | 79.8 | 53.4 | **46.4** | **68.0** | **56.4** | **37.2** |
| w/o $\mathcal{L}_{conf}$ | 84.4 | **88.8** | **84.6** | 23.8 | 54.2 | 38.0 | **84.0** | 53.8 | 33.9 | 66.0 | 54.8 | 31.4 |
| w/o $\mathcal{L}_{gate}$ | 86.0 | 88.4 | 80.6 | **27.4** | 56.6 | 37.6 | 83.4 | 44.8 | 17.9 | 61.0 | **56.4** | 33.0 |
| w/o $\mathcal{L}_{spar}$ | 84.6 | 87.6 | 80.2 | 26.6 | 54.4 | 38.2 | 82.6 | 38.0 | **46.4** | 66.0 | 52.4 | 35.2 |
| w/o $\alpha(x)$ | 68.2 | 80.4 | 47.6 | 21.4 | 52.0 | 30.2 | 72.0 | 49.2 | 39.3 | 57.0 | 52.4 | 33.2 |
| w/o $\gamma(x)$ | 64.8 | 82.2 | 49.2 | 21.0 | 54.8 | 29.6 | 73.0 | **57.4** | 39.3 | 56.0 | 52.8 | 33.0 |

heads. Both ablations cause consistent drops, showing that query-conditioned allocation is crucial: uniform $\alpha$ or $\gamma$ erases direction- and head-specific selectivity that underpins effective routing.

## 5 ANALYSES

### 5.1 ICR EXHIBITS INTERPRETABLE EFFECTS.

Though ICR modulates zero-shot inference in the attention space, its effects are interpretable. Probing next-token distributions across ID and OOD datasets reveals systematic vocabulary-level shifts that remain stable across datasets. Specifically, ICR consistently upweights tokens linked to reasoning-oriented structures such as *'capture'*, *'connections'*, and *'signs'*, rather than task-specific label words. Full method details and the top-50 ranked token list are provided in Appendix H.

### 5.2 ALIGNED AND DIVERSE DOMAIN DISTRIBUTIONS MATTER.

We study the impact of domain distribution in PIDs extraction and router training by varying the extraction and training data. Table 5 compares three configurations: (i) MATCHED-3: both extraction and training on {AGNews, SST-2, TREC}; (ii) MISMATCHED: extraction on {AGNews, SST-2, TREC} with {CSQA, PIQA} additionally included during training; (iii) MATCHED-5: extraction and training on all five datasets. Two key findings emerge. (1) Enlarging the training pool without aligning the extraction (MISMATCHED) degrades performance in most cases, as the router receives conflicting supervision signals that distort the extracted ICL patterns. (2) Jointly expanding both extraction and training (MATCHED-5) yields clear gains on OOD tasks, suggesting that the extracted ICL pattern becomes more generalizable (providing empirical support for our claim in Sec. 2.3). It also improves performance on ID tasks that appear unrelated to the added datasets (e.g., AGNews, TREC). This indicates that heterogeneous domains provide complementary ICL cues, enabling cross-task transfer and mutual reinforcement across domains.

### 5.3 ICR HIERARCHICALLY INTERNALIZES ICL DYNAMICS OF LLMs

In this section, we present a hierarchical importance analysis, spanning layers, heads, and PIDs, which progresses from coarse to fine granularity. This reveals how ICR adaptively composes ICL patterns across tasks and internalizes them at multiple levels of abstraction.

**Layer** We quantify per-layer contribution by combining two router signals: the mean head-level gate strength and the averaged weights in the routing vectors ($\alpha$) across $r$ directions. For each input, both streams are min–max normalized across layers, multiplied to form a layer-importance profile, and renormalized to sum to one. We then report dataset-level means restricted to the intervened layers. Figure 4 **Left** plots results for six representative datasets spanning ID, near-OOD, and far-OOD groups. The curves show that a few hub layers (notably 23 and 26) consistently dominate, suggesting that ICR identifies shared structural anchors for routing. Moreover, semantically related datasets (e.g., SST-5/MR, CB/MRPC) exhibit nearly parallel profiles, indicating that ICR adaptively reweights layers in a task-aware yet structurally consistent manner. A more detailed analysis with figures covering all 12 datasets is provided in Appendix I.

**Head** For each dataset, we record the gate values of all heads across layers for every zero-shot input and average them to obtain per-head importance scores. The head with the highest average value in each layer is selected as the Top-1 head, producing a routing sequence per dataset. We analyze six representative datasets from in-domain, near OOD, and far OOD groups, and visualize their routing sequences with a radar plot (Figure 4 **Middle**). The consensus hubs, marked with green stars, reveal that certain heads dominate the ICR process (e.g., head 26 in layer 22, head 21 in layer 23). In contrast, some layers exhibit task-specific divergence, where different tasks rely on different

Table 5: PIDs extraction and training with different domain combinations.

| Method | AG | SST-2 | TREC | CSQA | PIQA | SST-5 | MR | MRPC | CB | COPA | CREAK | AI2SciE |
|--------|-----|-------|------|------|------|-------|-----|------|-----|------|-------|---------|
| MATCHED-3 | 86.4 | **87.6** | 79.6 | 21.4 | 51.0 | 35.6 | **80.2** | 60.4 | 37.5 | 57.0 | 52.8 | 34.2 |
| MISMATCHED | 65.0 | 82.8 | 63.6 | 23.4 | 54.6 | 29.8 | 76.4 | **64.0** | 32.1 | 65.0 | 53.6 | 30.8 |
| MATCHED-5 | **86.6** | 86.4 | **83.8** | **24.8** | **57.0** | **38.6** | 79.8 | 53.4 | **46.4** | **68.0** | **56.4** | **37.2** |

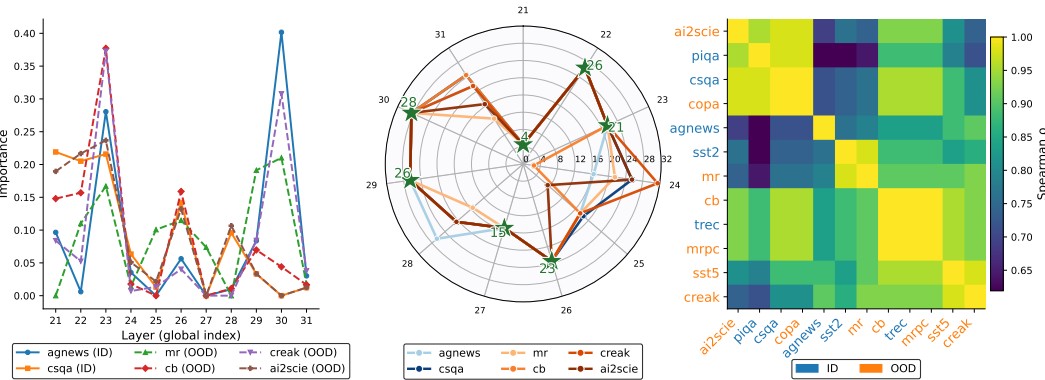

Figure 4: **Left**: Layer-importance visualization under ICR. **Middle**: Visualization of top-1 head in each layer, with rings for heads, spokes for layers (starting at layer 21), and green stars marking consensus heads (numbers denote head indices). **Right**: Correlation of per-dataset PID importance.

heads (e.g., at layer 28 the six tasks split across three heads, indicating three routing modes). These results show that ICR identifies shared hub heads while flexibly adapting routing in non-hub layers.

**PIDs** We estimate per-dataset PID importance by combining the absolute weights in $\alpha$ with the average head-gate strength in each layer, and then averaging these weighted values across layers. For each dataset, this yields a vector whose entries correspond to the importance of individual PIDs. Pairwise Spearman correlations of these vectors are calculated and clustered (Figure 4 **Right**). The results show that ICR flexibly combines and routes along different ICL directions: for example, MR aligns more with SST-2/TREC, while AI2SCIE and COPA correlate more with CSQA/PIQA, reflecting a greater dependence on reasoning-oriented patterns than sentiment- or classification-oriented patterns. This differentiated behavior confirms that our attention routing-based design can dynamically select and exploit relevant ICL directions, enabling adaptation across diverse OOD scenarios. These results demonstrate the deep alignment between ICR and the attention mechanisms, which can benefit continually evolving transformer-based models.

## 6 RELATED WORK

**Implicit In-context Learning.** To better understand and exploit ICL, prior work has emphasized the role of MHA. Building on these insights, researchers have proposed implicit ICL, which converts ICDs into vectors injected into LLM activations, typically within MHA (Merullo et al., 2023). Task Vectors (Hendel et al., 2023) are extracted from specific layers, while Function Vectors (Todd et al., 2023) come from attention heads critical to ICL; both are applied during zero-shot inference to provide task-relevant knowledge. Liu et al. (2023) modeled ICDs as shifts on MHA outputs and introduced the in-context vector, while Peng et al. (2024); Jiang et al. (2025); Li et al. (2025a) developed training strategies to enhance vector expressiveness. Although these methods alleviate the latency and instability of token-level ICDs (Chen et al., 2022; Xiang et al., 2024), their limited theoretical grounding in attention restricts generalization. Our approach, ICR, addresses this gap and opens a new direction for implicit ICL. Additional related works are introduced in Appendix J.

## 7 CONCLUSION

We introduce In-Context Routing (ICR), a query-conditioned framework that extracts and exploits generalizable ICL patterns within the MHA module of LLMs. Extensive experiments demonstrate that ICR delivers robust performance across diverse ID and OOD tasks. Moreover, it requires only a single round of training and transfers to new tasks without additional retrieval or retraining. By operationalizing the mechanism of ICL within the implicit ICL paradigm, ICR improves both effec-

tiveness and efficiency and further extends the benefits of ICL to tasks without labeled examples. ICR provides valuable insights for reshaping zero-shot inference in the next generation of LLMs.

## REPRODUCIBILITY STATEMENT

The LLMs adopted in this study are presented in Sec.4.1. The training procedures with full hyper-parameter settings are reported in Appendix D.2, and details of the datasets used in this study are provided in Appendix D.3.1. Due to our institution's privacy policy and the requirements of double blind review, we will release all code used for data reprocessing and for conducting experiments upon the publication of the paper. The code will be distributed under a license that permits free use for research purposes.

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

# A SUPPLEMENTARY THEORETICAL ANALYSIS

## A.1 KERNEL VIEW OF ATTENTION ROUTING

Self-attention can be viewed as a kernel machine, where the dot-product $q^\top k$ defines a *linear kernel* $K_0(q,k) = q^\top k$. From this perspective, attention routing does not merely add a bias to the logits, but reparameterizes the kernel itself. Formally, let $Q_{\mathbf{zs}}^l, K_{\mathbf{zs}}^l \in \mathbb{R}^{T \times d}$ be the layer-level projections during zero-shot inference. We define a reparameterized kernel function

$$K_\alpha^l(q,k) = q^\top M^l(\alpha^l) k, \tag{15}$$

where the reparameterization matrix is

$$M^l(\alpha^l) = I_d + U_q^l \operatorname{diag}(\alpha^l) U_k^{l\top}. \tag{16}$$

Here $U_q^l, U_k^l \in \mathbb{R}^{d \times r}$ are the PID bases and $\alpha^l \in \mathbb{R}^r$ is the routing vector. The resulting correction is

$$\Delta A^l = Q_{\mathbf{zs}}^l M^l(\alpha^l) K_{\mathbf{zs}}^l - Q_{\mathbf{zs}}^l K_{\mathbf{zs}}^l,$$

which is then broadcast to heads to produce

$$\tilde{A}^{l,h} = A^{l,h} + \Delta A^l.$$

This kernel view shows that attention routing replaces the fixed linear kernel with a reparameterized kernel whose deviation from $K_0$ is low-rank, since $\operatorname{rank}(M^l(\alpha^l) - I) \leq r$. The modification is structural, as it is confined to PID directions.

## A.2 SPIKED COVARIANCE MODEL

The *spiked covariance model* (Johnstone, 2001) is a widely studied framework in random matrix theory and high-dimensional statistics. It assumes that the population covariance matrix $\Sigma \in \mathbb{R}^{d \times d}$ can be decomposed into an isotropic noise component plus a small number of low-rank "spikes":

$$\Sigma = \sum_{i=1}^{r} \theta_i u_i u_i^\top + \sigma^2 I_d, \tag{17}$$

where $\sigma^2 I_d$ represents homogeneous noise, $u_i \in \mathbb{R}^d$ are orthonormal eigen-directions corresponding to signal components, and $\theta_i$ are the spike strengths (eigenvalues above the noise level). In this setting, most eigenvalues of $\Sigma$ concentrate around $\sigma^2$, while a few leading eigenvalues (the spikes) separate from the bulk, capturing the essential low-dimensional structure of the data. This model provides the foundation for our mixed spiked formulation, where we separate shared low-dimensional attention structures from domain-specific variations to analyze in-context reasoning signals across datasets.

## A.3 FORMAL ANALYSIS OF POOLED PCA

We provide a high-level analysis supporting the claim in Sec. 2.3 that pooled PCA over multiple domains can better recover the general ICL pattern. Our argument is based on the classical Davis–Kahan $\sin\Theta$ theorem (Davis & Kahan, 1970), which bounds the deviation between the estimated and true subspaces under perturbations. Let $\widehat{U}_q$ be the top-$r$ eigenspace of the pooled covariance $\widehat{\Sigma}_Q$, and let $S_q$ denote the ground-truth shared subspace. Then

$$\sin\Theta\big(\operatorname{span}(\widehat{U}_q), \operatorname{span}(S_q)\big) \lesssim \frac{\tilde{O}(N^{-1/2}) + \rho_D}{\operatorname{gap}_Q}, \tag{18}$$

where $\operatorname{gap}_Q$ is the eigengap separating the shared spikes from the bulk spectrum. Here, $\sin\Theta(U, V)$ denotes the operator norm of the sine of the canonical angles between subspaces $U$ and $V$. The numerator of the bound contains two sources of error: the $\tilde{O}(N^{-1/2})$ term from finite-sample noise and the residual $\rho_D$ from domain-specific variations. Both decrease with larger $N$ and $D$: increasing $N$ reduces sampling fluctuations, while increasing $D$ averages out heterogeneous domain-specific directions.

At the same time, the denominator $\operatorname{gap}_Q$ becomes larger as $N$ and $D$ grow. With more samples, the leading eigenvalues of the shared component are estimated more accurately, and with more domains,

domain-specific contributions cancel out, making the shared spikes stand out more prominently from the bulk.

Together, these effects tighten the Davis–Kahan bound: the numerator shrinks while the denominator enlarges, so the subspace distance $\sin\Theta(\widehat{U}_q, S_q)$ decreases. Consequently, pooled PCA on multi-domain ICL bases becomes increasingly reliable for recovering the shared subspace $S_q$.

### A.4 PERTURBATION ANALYSIS OF OOD STABILITY

We continue our analysis by showing that the shared ICL subspace recovered by pooled PCA is not only stable under test-time distribution shifts but also becomes more accurate for out-of-distribution (OOD) generalization as the number of domains increases. Specifically, we model OOD shifts in the query/key statistics as additive perturbations to the pooled covariances:

$$\widehat{\Sigma}'_Q = \widehat{\Sigma}_Q + \Delta_Q, \qquad \widehat{\Sigma}'_K = \widehat{\Sigma}_K + \Delta_K,$$

where $\|\Delta_Q\|_{\mathrm{op}}, \|\Delta_K\|_{\mathrm{op}} \leq \epsilon$ capture bounded changes in second-order statistics.

Let $U_q$ be the top-$r$ eigenspace of $\widehat{\Sigma}_Q$, and $\widehat{U}_q$ be the corresponding eigenspace of the perturbed matrix $\widehat{\Sigma}'_Q$. The Davis–Kahan $\sin\Theta$ theorem (Davis & Kahan, 1970) gives the bound:

$$\sin\Theta\big(\mathrm{span}(\widehat{U}_q), \mathrm{span}(U_q)\big) \;\leq\; \frac{\|\Delta_Q\|_{\mathrm{op}}}{\mathrm{gap}_Q} \tag{19}$$

Thus, the subspace stability depends on the relative size of the perturbation versus the eigengap. An identical argument applies to $U_k$.

Importantly, pooling across multiple domains helps enlarge $\mathrm{gap}_Q$ by amplifying the shared signal while averaging out domain-specific variations (see Sec.2.3). This increases the separation between the top-$r$ eigenvalues and the noise floor, which tightens the Davis–Kahan bound and ensures that the perturbed subspace $\widehat{U}_q$ remains closer to the in-domain subspace $U_q$ under test-time shifts. Together, these explain why increasing the number of training domains leads to more reliable OOD routing in practice.

## B  CHALLENGES OF VECTOR-BASED IMPLICIT ICL

Although vector-based methods can reproduce certain *input-output statistics* of ICDs and enable efficient ICL without token-level ICDs, they suffer from two fundamental challenges.

1.**Weak theoretical grounding limits scalability.** Vector-based methods convert certain explicit ICDs into free-form residual biases of a specific model **without** structural connections to the query/key space, which makes them relatively black-box and detached from the theoretical framework of MHA. Thus, these methods witness large performance fluctuations when transferred across architectures. Moreover, incorporating new knowledge into these vectors or resizing them to fit novel models requires curated training, and the results of such training can also be unstable.

2.**Post-hoc residual steering limits generalization.** Vector-based implicit ICL intervenes only after attention aggregation, injecting additive shifts into the MHA output. Such post-hoc adjustments lack structural control: the resulting representations are often entangled with task-specific content, limiting their ability to transfer beyond the training task. Since the underlying attention logits $\mathbf{A}^{l,h}$, which more fundamentally encode ICL patterns, remain unaffected, the model tends to mimic ICL by fitting specific feature patterns rather than developing the attention dynamics needed to exploit context. This design inherits the potential attention deficits in explicit ICL (Lee et al., 2023), while also lacking the adaptability necessary for multi-task or OOD scenarios.

## C  ICR PSEUDOCODE

ICR consists of two key phases: PIDs extraction and router training. Algorithm 1 presents the pseudocode for multi-task query/key representation collection and the subsequent PIDs extraction, while Algorithm 2 illustrates the core mechanism and training procedure of ICR.

---

**Algorithm 1:** Collecting PIDs $U_q, U_k$ across multiple domains

---

**Input:** Model $M$, datasets $\{D_1, \ldots, D_N\}$ with $M_n$ prompts each, target layers $L$, PCA rank $r$
**Output:** $U_q^l, U_k^l$ for each $l \in L$

**1 foreach** $l \in L$ **do**
  **2**   $Q_{\text{pool}}[l] \leftarrow \emptyset, \quad K_{\text{pool}}[l] \leftarrow \emptyset$
**3 foreach** *dataset* $D_n$ **do**
  **4**   **for** $i \leftarrow 1$ **to** $M_n$ **do**
    **5**   $p \leftarrow \text{GenerateFewShotPrompt}(D_n)$;
    **6**   Run $M(p)$ with Q/K hooks;
    **7**   **foreach** $l \in L$ **do**
      **8**   $Q_{\text{last}}^l \leftarrow \text{Concat}_{h=1}^H Q_{l,h}[t_{\text{last}}]$;
      **9**   $K_{\text{last}}^l \leftarrow \text{Concat}_{h=1}^H K_{l,h}[t_{\text{last}}]$;
      **10**   Append $Q_{\text{last}}^l$ to $Q_{\text{pool}}[l]$;
      **11**   Append $K_{\text{last}}^l$ to $K_{\text{pool}}[l]$;

**12 foreach** $l \in L$ **do**
  **13**   $Q \leftarrow \text{Concat}(Q_{\text{pool}}[l])$;
  **14**   $K \leftarrow \text{Concat}(K_{\text{pool}}[l])$;
  **15**   $U_q^l \leftarrow \text{Top-}r\ \text{PCA}(Q)$;
  **16**   $U_k^l \leftarrow \text{Top-}r\ \text{PCA}(K)$;
  **17**   Save $U_q^l, U_k^l$;

---

# D   EXPERIMENTAL SETUP

## D.1   COLLECTION DETAILS

To construct the ICL bases, we collect 10k examples from AGNEWS and 5k examples each from SST-2, TREC, CSQA, and PIQA. This allocation is motivated by the complementary characteristics of these datasets: AGNEWS focuses on topic-level categorization that captures broad semantic content, SST-2 and TREC emphasize sentence-level classification with a sharper focus on specific linguistic distinctions, while CSQA and PIQA represent QA-style tasks that require more reasoning-oriented processing. Overall, this balanced collection is designed to provide approximately uniform coverage of semantic, classification, and reasoning patterns.

## D.2   TRAINING DETAILS

Optimization uses AdamW (lr $1 \times 10^{-4}$, batch size 4) for 2 epochs with gradient clipping (1.0) and PIDs rank $r{=}8$. The training objective combines cross-entropy with a confidence-improvement term ($\lambda_{\text{conf}}{=}0.01$), an $\ell_1$ sparsity penalty on routing vectors ($\lambda_{\text{spar}}{=}10^{-3}$), and a gate sparsity term ($\lambda_{\text{gate}}{=}0.02$). To stabilize training, we employ two simple schedules: (i) a late-layer weighting scheme that increases sparsity strength toward the late layers (up to 3.0) ($w^l$ in Eq.13), and (ii) a cosine annealing of the routing scale $\alpha$ across epochs (from 1.0 to 0.8). Inputs to both the encoder and the LLM are truncated to 512 tokens. All runs use a single V100 GPU under deterministic settings (seed 42; TF32 and non-deterministic SDPA disabled).

## D.3   EVALUATION DETAILS

Predictions follow a unified next-token scoring protocol: each answer option is mapped to the variant that tokenizes into a single token, and the prediction is taken as the $\arg\max$ over the logits at the next position restricted to these candidate ids. When ICR is enabled, the router is conditioned on a mean-pooled MiniLM sentence embedding, while the backbone remains frozen.

### D.3.1   DATASETS

**In-Domain** We treat the five datasets used for cross-domain collection and router training as in-domain: AGNews, SST-2, TREC, CSQA, and PIQA. **AGNews** provides large-scale topic classification over news articles spanning four domains. **SST-2** evaluates binary sentiment classification on movie reviews, emphasizing subtle polarity cues. **TREC** focuses on open-domain question classification into several semantic types. **CSQA** targets commonsense reasoning through multiple-choice

---

**Algorithm 2:** In-Context Routing (ICR) training

---

**Input:** Frozen backbone $M$ with $L$ layers and $H$ heads per layer;
Frozen encoder $E$; Router MLP with parameters $\theta = (\theta_\alpha, \theta_\gamma)$;
Subspaces $\{U_q^l, U_k^l\}_{l=1}^L$;
Late-layer set $\mathcal{L}_{\text{late}} = \{ \frac{2L}{3} + 1, \ldots, L \}$;
Datasets $\{D_1, \ldots, D_N\}$, where each sample is $(x, y, d)$ with input $x$, label $y$, dataset index $d$;
Optimizer $\text{Opt}(\theta)$
**Output:** Trained router parameters $\theta$

**1 while** *not converged* **do**

    // sample a minibatch from the union of datasets

**2**     $\mathcal{B} \leftarrow \{(x_i, y_i, d_i)\}_{i=1}^B$ ;

**3**     **for** $i = 1$ **to** $B$ **do**

**4**         $z_i \leftarrow E(x_i)$ ;             // pooled representation of the query

**5**         $(\alpha_i, \gamma_i) \leftarrow \text{RouterMLP}_\theta(z_i)$ ;     // $\alpha_i[l,:] \in \mathbb{R}^r$, $\gamma_i[l,h] \in \mathbb{R}$

**6**         **for** $l = 1$ **to** $L$ **do**

**7**             **if** $l \notin \mathcal{L}_{\text{late}}$ **then continue**;

**8**             $Q \leftarrow W_q^l h^l(x_i), \quad K \leftarrow W_k^l h^l(x_i), \quad V \leftarrow W_v^l h^l(x_i)$;

**9**             $Z_q \leftarrow Q\, U_q^l \in \mathbb{R}^{T \times r}, \quad Z_k \leftarrow K\, U_k^l \in \mathbb{R}^{T \times r}$;

**10**            $B_{\text{shared}} \leftarrow \text{einsum}(Z_q,\, \alpha_i[l,:],\, Z_k) \in \mathbb{R}^{T \times T}$;

**11**            **for** $h = 1$ **to** $H$ **do**

**12**                $B_{\text{head}}^{(h)} \leftarrow \gamma_i[l,h] \cdot B_{\text{shared}}$ ;

**13**                $S^{(h)} \leftarrow \frac{Q^{(h)} K^{(h)\top}}{\sqrt{d_k}} + B_{\text{head}}^{(h)}$ ;

**14**                $A^{(h)} \leftarrow \text{softmax}(S^{(h)})$ ;

**15**                $O^{(h)} \leftarrow A^{(h)} V^{(h)}$ ;

           // replace layer-$l$ attention output

**16**            $\tilde{O}^l \leftarrow \text{Concat}_{h=1}^H O^{(h)}\, W^O$ ;

        // obtain task logits at the last token

**17**         $\ell_i \leftarrow M(x_i)\big|_{\text{last-token}}$ ;

    // training loss

**18**     $L_{\text{task}} \leftarrow \{\ell_i^{\text{ICR}}, y_i\}$ ;

**19**     $L_{\text{conf}} \leftarrow \{\ell_i^{\text{ICR}}, \ell_i^{\text{zs}}\}$ ;

**20**     $L_{\alpha\text{-spar}} \leftarrow \{\alpha\}$ ;

**21**     $L_{\gamma\text{-spar}} \leftarrow \{\gamma\}$ ;

**22**     $L \leftarrow L_{\text{task}} + \lambda_{\text{conf}} L_{\text{conf}} + \lambda_\alpha L_{\alpha\text{-spar}} + \lambda_\gamma L_{\gamma\text{-spar}}$ // router update

**23**     $\text{Opt.zero\_grad}(); \quad \nabla_\theta L; \quad \text{Opt.step}()$

---

questions grounded in everyday knowledge. **PIQA** assesses physical knowledge by requiring plausibility judgments over everyday actions.

**Out-of-Domain** For out-of-domain (OOD) evaluation, we consider seven representative datasets that are disjoint from the collection and training sources. We divide them into *near OOD* and *far OOD* groups depending on their proximity to the training tasks in terms of domain, label space, and input format.

Firstly we articulate a clear framework for how we distinguish near-OOD and far-OOD. In the context of ICL generalization, we consider three key axes: (i) the compatibility of the label ontology with the ID family, (ii) the similarity of the required operation or reasoning structure, and (iii) the degree of semantic and domain shift. Under this framework, near-OOD tasks are those that deviate from ID along at most one of these axes while preserving the overall ICL structure. SST-5, MR, and MRPC fall into this category. SST-5 extends SST-2 from a binary polarity to a five-point sentiment scale, so the label ontology changes in granularity but remains sentiment-based, while the operation type and domain are essentially identical. MR keeps the same binary sentiment ontology as SST-2 and the same sentence-level classification operation, but shifts the underlying review corpus, so the main change is semantic and domain shift. MRPC, although cast as sentence-pair paraphrase,

Table 6: Datasets, task types, and prompt templates used in ICR.

| Dataset | Task Type | Template |
|---------|-----------|----------|
| AGNews | Topic classification | News: {text}; Type: [World, Sports, Business, Technology] |
| SST-2 | Sentiment (binary) | Review: {text}; Sentiment: [negative, positive] |
| TREC | Question type classification | Question: {text}; Answer Type: [Abbreviation, Entity, Description, Person, Location, Number] |
| CSQA | Commonsense MCQ (5-class) | Question: {question}; A. {optA}; B. {optB}; C. {optC}; D. {optD}; E. {optE}; Answer (A/B/C/D/E); Options: [A, B, C, D, E] |
| PIQA | Physical commonsense (2-choice) | Goal: {goal}; A. {optA}; B. {optB}; Answer (A/B); Options: [A, B] |
| SST-5 | Sentiment (5-class) | Sentence: {text}; Sentiment: [terrible, negative, neutral, positive, great] |
| MR | Movie Review (binary) | Review: {text}; Sentiment: [negative, positive] |
| MRPC | Paraphrase | {pair}; A. Paraphrase; B. Not paraphrase; Answer (A/B); Options: [A, B] |
| CB | NLI (3-class) | {pair}; A. Entailment; B. Contradiction; C. Neutral; Answer (A/B/C); Options: [A, B, C] |
| CREAK | Claim verification | Claim: {claim}; Label: yes / no; Options: [yes, no] |
| COPA | Causal reasoning (2-choice) | {context}; A. {optA}; B. {optB}; Answer (A/B); Options: [A, B] |
| AI2SciE | Science MCQ (K-choice) | Question: {question}; A. {optA}; B. {optB}; C. {optC}; D. {optD}; E. {optE}; F. {optF}; G. {optG}; H. {optH}; Answer (A/B/C/...); Options: [A, B, C, D, E, F, G, H] |

still uses a binary decision geometry that is compatible with SST-2 and the A/B decision format of PIQA, and the reasoning required is largely surface-level alignment such as lexical and syntactic rewrites. In this sense it introduces a mild change in operation type but remains close to ID along label ontology and general language style. By contrast, far-OOD tasks shift along multiple axes at once. CB, COPA, CREAK, and AI2SciE all introduce new label inventories and reasoning structures that are not present in any ID dataset, together with nontrivial semantic shifts. CB is a three-way NLI task with labels entailment, contradiction, and neutral, which do not align with any ID label space, and it requires directional inference from premise to hypothesis. COPA formulates explicit causal reasoning over alternatives, which differs from the recognition-style decisions in ID and entails a different type of relational reasoning. CREAK focuses on claim verification, relying on world knowledge and on reasoning about when seemingly plausible statements fail in specific cases, rather than on shallow sentence-level judgments. AI2SciE requires scientific explanatory reasoning over domain-specific content that is absent from the ID datasets. Together, these shifts in label ontology, operation type, and semantics alter the effective ICL geometry in more than one dimension.

*Near OOD.* **SST-5** evaluates fine-grained sentiment prediction beyond the binary labels seen in training, requiring models to calibrate over a five-class space. **MR** further tests domain transfer by shifting sentiment analysis to the movie-review domain. Finally, **MRPC** evaluates robustness under input format shift, where the model must generalize from single-sentence classification to sentence-pair paraphrase detection. These tasks remain relatively close to the training distribution (sentiment or classification-style tasks) but introduce moderate shifts in label granularity, domain, or input structure.

*Far OOD.* In contrast, **CommitmentBank (CB)** stresses generalization under shifts in semantic judgment criteria, where decisions hinge on subtle pragmatic or syntactic cues absent from typical training tasks. **COPA** introduces a pairwise choice format grounded in causal reasoning. **CREAK** evaluates plausibility judgments in commonsense relational contexts. Finally, **AI2SciE** requires elementary science question answering, representing a shift toward multi-hop reasoning. These

datasets constitute far OOD scenarios, as they deviate more substantially from the training distribution in both task format and reasoning requirements.

Taken together, the near and far OOD sets cover complementary axes of generalization, ranging from finer-grained variants of familiar tasks to entirely novel reasoning paradigms, thus providing a comprehensive testbed for out-of-domain robustness. On these datasets we report comparisons only with zero-shot and few-shot prompting, since current vector- or retrieval-based methods require labeled in-domain ICDs and are not directly applicable.

**Templates**   The datasets used for extraction, training, and evaluation are listed in Table 6, along with their task types and templates. For in-domain datasets, the templates serve a dual role: they are applied when constructing ICL prompts prior to collecting query/key representations for PCA-based PIDs extraction, and again during evaluation. For out-of-domain datasets, the templates are employed only for evaluation.

### D.3.2 PRELIMINARY EXPERIMENT SETUP

For the preliminary cross-task ICL experiments in Section 1 (Figure 1), the inference-time model is Llama-2-7B, and all prompts follow the template shown in Table 6. Each experiment uses a total of 16 in-context demonstrations. In the single-source setting, we sample 16 demonstrations without replacement from the training split of a single source task. In the cross-task setting, we sample 8 demonstrations from each of two source tasks, concatenate them, and uniformly shuffle their order before inserting them into the prompt. In both settings, demonstrations from any dataset are selected using label-balanced sampling.

For evaluation, each target task is assessed on a subset of 500 test instances, and we report accuracy averaged over 5 independent seeds. Decoding is performed using greedy search.

### D.3.3 BASELINES

For in-domain evaluation, we compare our method against several representative vector-based implicit ICL baselines, including Task Vector (TV), Function Vector (FV), In-Context Vector (ICV), ELICIT, Iterative Vectors (IV), Implicit ICL (I2CL), Learnable In-context VEctor (LIVE), and M2IV, in addition to standard zero-shot and few-shot prompting. For out-of-domain evaluation, we select three methods that involve calibration or training with data: I2CL, LIVE, and M$^2$IV, as other training-free methods **cannot** be applied to OOD tasks. For methods requiring training, we follow the original setups and conduct a hyperparameter search to achieve the best performance. The details of the baselines are as follows:

- Task Vector (TV): TV frames ICL as compressing the demonstrations into a single task vector that encodes the task rule. This vector is then patched into the transformer's intermediate layers during the query's forward pass, steering the model's prediction without direct access to the demonstrations.

- Function Vector (FV): FVs identify a small set of causal attention heads that transport a compact vector representation of the demonstrated task during ICL. By extracting this function vector and inserting it into the hidden states of new contexts, the model can execute the task in zero-shot or natural text settings. The approach shows that LLMs internally encode portable and composable task representations.

- In-Context Vector (ICV): ICVs recast ICL by extracting a single vector from the latent states of demonstration examples, which summarizes the task. At inference, this vector is added to the hidden states of all layers during the query's forward pass. This approach improves controllability, reduces context length, and supports vector arithmetic for combining tasks.

- ELICIT: ELICIT introduces a modular framework that builds a capability library of task vectors extracted from in-context learning prompts. At inference, a retrieval module dynamically selects and injects relevant task vectors into the model's hidden states, enabling it to reuse learned capabilities without extra tokens or fine-tuning.

- Iterative Vectors (IV): IVs enhance ICL by extracting activation-based meta-gradients, the differences between activations with and without demonstrations, and refining them

Table 7: Baseline comparison across benchmarks. *For ID datasets, few-shot uses 5-shot balanced sampling per class. For OOD datasets, we adopt multi-task few-shot prompting where each ID dataset provides 3-shot ICDs. Under **Overall**, *Average* is the mean accuracy across all datasets, and *Collapse* counts datasets where a method underperforms the zero-shot baseline.

| Method | In-Domain (ID) | | | | | Near OOD | | | Far OOD | | | | Overall | |
|---|---|---|---|---|---|---|---|---|---|---|---|---|---|---|
| | AG | SST-2 | TREC | CSQA | PIQA | SST-5 | MR | MRPC | CB | COPA | CREAK | AI2SciE | Average | Collapse |
| | | | | | | | Llama3.1-8B | | | | | | | |
| Zero-shot | 70.0 | 87.8 | 49.0 | 65.0 | 62.6 | 27.6 | 82.2 | 68.8 | 41.1 | 65.0 | 53.6 | 78.4 | 62.6 | – |
| Few-shot* | 88.2 | 91.4 | 57.4 | 72.8 | 70.4 | 42.2 | 91.8 | 72.4 | 51.4 | 63.0 | 50.8 | 89.6 | 70.1 | 2 |
| I2CL | 79.8 | 86.4 | 63.8 | 66.2 | 62.0 | 30.8 | 82.0 | 64.8 | 40.6 | 61.2 | 46.8 | 61.4 | 62.2 | 8 |
| LIVE | 82.6 | 87.8 | 66.0 | 66.8 | 61.4 | 32.4 | 78.6 | 69.0 | 41.8 | 58.8 | 51.0 | 65.2 | 63.5 | 5 |
| M²IV | 83.4 | 88.2 | 64.8 | 67.2 | 64.8 | 35.0 | 81.8 | 67.8 | 42.6 | 60.8 | 49.8 | 67.6 | 64.5 | 5 |
| **ICR** | **85.2** | **88.6** | **76.8** | 66.6 | **66.4** | **36.6** | **83.6** | **69.4** | **42.9** | **67.0** | **54.6** | **82.6** | **68.4** | **0** |

Table 8: Comparison of ICR and LoRA. **Param.** denotes the number of trainable parameters relative to ICR (with ICR set as ×1.0).

| Method | In-Domain (ID) | | | | | Near OOD | | | Far OOD | | | | Overall | |
|---|---|---|---|---|---|---|---|---|---|---|---|---|---|---|
| | AG | SST-2 | TREC | CSQA | PIQA | SST-5 | MR | MRPC | CB | COPA | CREAK | AI2SciE | Average | Param. |
| | | | | | | | Qwen2.5-7B | | | | | | | |
| LoRA | **83.6** | **93.2** | **71.6** | **84.0** | **84.2** | 40.8 | 88.5 | **73.2** | 83.0 | 92.6 | 74.6 | 91.5 | 80.1 | ×2.1 |
| ICR | 80.4 | 91.0 | 70.6 | 82.0 | 82.6 | **41.4** | **89.4** | **73.2** | **84.6** | **95.0** | **79.2** | **93.2** | **80.2** | ×1.0 |
| | | | | | | | Llama-3.1-8B | | | | | | | |
| LoRA | **86.8** | **90.4** | **77.2** | **67.2** | 65.8 | **37.4** | 83.0 | 69.0 | 40.0 | 65.4 | 52.6 | 79.8 | 67.9 | ×2.8 |
| ICR | 85.2 | 88.6 | 76.8 | 66.6 | **66.4** | 36.6 | **83.6** | **69.4** | **42.9** | **67.0** | **54.6** | **82.6** | **68.4** | ×1.0 |

through an iterative process. These vectors are then injected back into the model's activations during inference, effectively simulating gradient updates without backpropagation.

- Implicit ICL (I2CL): I2CL extracts vectors from each ICD and aggregates them into a unified context vector. During inference, it injects a linear combination of this context vector and the query activations into each layer's residual streams to simulate the effect of ICL. Additionally, I2CL employs a noisy self-calibration step to optimize the layer-wise fusion coefficients.

- Learnable In-context VEctor (LIVE): LIVE distills task information from ICDs into a set of learnable vectors. During training, it aligns the model's outputs using ICDs with those using LIVE, and at inference, the learned vectors are added to each layer's MHA outputs to simulate the effect of ICDs.

- M²IV: M²IV assigns learnable vectors and weight factors to both the MHA and MLP branches at each layer of an LVLM. During training, it uses a self-distillation framework with mimicry, synergistic, and supervised losses to align with Vanilla ICL outputs. At inference, the trained vectors are injected into residual streams to emulate n-shot ICL without explicit ICDs.

# E    ADDITIONAL RESULTS

## E.1    RESULTS ON LLAMA3.1-8B

Table 7 presents additional results comparing ICR with zero-shot, few-shot, and baseline methods on Llama3.1-8B. Overall, ICR approaches and sometimes surpasses few-shot performance, while consistently outperforming other task-specific implicit ICL baselines in both accuracy and stability. Notably, ICR shows no collapses below zero-shot performance on any OOD task, outperforming multi-task few-shot prompting and all other baselines. This reinforces our conclusions in Sec. 4.2.

## E.2    COMPARISON WITH LORA

We further compare ICR with LoRA in Table 8. The LoRA module is applied to the token classification head of the last layer with rank 32. For training, we use the same number of few-shot examples as those contained in an ICL prompt during the construction of ICL bases, drawn from five in-domain datasets. Although LoRA requires 2–3× more trainable parameters than ICR, it achieves slightly weaker overall performance. Moreover, ICR exhibits clear advantages in OOD settings, which shows its better generalizability and efficiency compared to the PEFT-based methods in few-shot scenarios.

Table 9: Baseline comparison across benchmarks on Qwen3-32B and Llama3.1-70B. [*]For ID datasets, few-shot uses 5-shot balanced sampling per class. For OOD datasets, we adopt multi-task few-shot prompting where each ID dataset provides 3-shot ICDs.

| Method | In-Domain (ID) | | | | | Near OOD | | | Far OOD | | | | Overall |
|---|---|---|---|---|---|---|---|---|---|---|---|---|---|
| | AG | SST-2 | TREC | CSQA | PIQA | SST-5 | MR | MRPC | CB | COPA | CREAK | AI2SciE | Average |
| Qwen3-32B | | | | | | | | | | | | | |
| Zero-shot | 69.8 | 83.0 | 51.2 | 76.0 | 78.0 | 46.0 | 95.2 | 75.0 | 91.1 | 96.0 | 51.8 | 85.8 | 74.9 |
| Few-shot[*] | 84.4 | 89.8 | 77.8 | 86.8 | 89.0 | 43.8 | 99.4 | 76.4 | 91.1 | 98.0 | 51.4 | 80.3 | 80.7 |
| FV | 74.6 | 82.6 | 58.2 | 75.2 | 63.4 | 35.6 | 93.4 | 74.8 | 85.2 | 93.8 | 47.0 | 79.4 | 71.9 |
| I2CL | 78.4 | 85.6 | 64.7 | 74.6 | 74.2 | 38.6 | 94.8 | 76.0 | 89.6 | 94.2 | 52.0 | 84.6 | 75.6 |
| **ICR** | **81.6** | **86.4** | **77.2** | **82.0** | **82.8** | **50.4** | **97.2** | **79.8** | **94.6** | **96.0** | **53.6** | **88.6** | **80.9** |
| Llama3.1-70B | | | | | | | | | | | | | |
| Zero-shot | 48.8 | 93.2 | 62.6 | 80.2 | 70.4 | 44.0 | 82.0 | 71.4 | 91.0 | 96.0 | 92.6 | 68.6 | 77.6 |
| Few-shot[*] | 70.8 | 91.0 | 68.0 | 83.2 | 88.6 | 46.4 | 85.2 | 78.6 | 92.9 | 97.0 | 88.0 | 96.2 | 82.2 |
| FV | 52.4 | 86.4 | 58.4 | 75.6 | 71.2 | 42.6 | 78.8 | 68.4 | 85.6 | 86.4 | 85.8 | 80.8 | 72.7 |
| I2CL | 62.6 | 88.8 | 63.0 | 73.8 | 75.4 | 46.8 | 77.6 | 70.0 | 90.2 | 89.0 | 88.0 | 84.4 | 75.8 |
| **ICR** | **66.4** | **93.8** | **66.0** | **82.4** | **83.2** | **48.4** | **86.8** | **80.2** | **93.4** | **92.0** | **93.2** | **92.0** | **81.5** |

### E.3 RESULTS ON MODELS WITH LARGER SCALE

To test the robustness of ICR on larger-scale models, we additionally report experiments on increased model sizes in Table 9. Across the models with increasing scale, ICR consistently outperforms the two residual-injection baselines. It approaches few-shot performance in in-domain settings and typically surpasses cross-task few-shot prompting in OOD settings, with only a few exceptions where the large-scale model is already very strong and results become slightly unstable. These trends further validate that ICR achieves stable transfer by avoiding reliance on noisy cross-task ICDs, and demonstrate its effectiveness across models of different scales.

## F EFFICIENCY ANALYSIS

To assess the efficiency of In-Context Routing (ICR), we benchmark it against baselines along two dimensions. Following Li et al. (2024), we report cached parameter size in Table 11. For ICR, the cached parameter is $2rdL$, as both $U_q$ and $U_k$ of the shape $d \times r$ must be stored in each layer. Although this appears larger than some baselines, $r$ is typically a small constant (e.g., 4–16), so the asymptotic complexity remains $\mathcal{O}(dL)$, on par with methods such as I2CL or LIVE. Moreover, since $r \ll M$ in few-shot settings, ICR still provides a far lighter memory footprint compared to explicit ICL.

We also report the average per-sample inference time over five in-domain datasets in Figure 5. The results show that ICR consistently requires less inference time than the 5-shot setting. More importantly, as the input length increases, the inference time of few-shot grows much faster than that of ICR. This demonstrates that ICR preserves the efficiency of implicit ICL, with the advantage becoming especially pronounced for longer contexts.

For a better understanding of offline computational cost of ICR, we provide an explicit comparison with I2CL, M2IV, and LIVE in Table 10. The cost is measured in NVIDIA V-100 GPU hours for representation collection and training/calibration. Because ICR extracts PIDs and trains the router using five in-domain datasets, and the baselines are not inherently designed for OOD scenarios, we report their offline cost on ID tasks only. Concretely, ICR performs a single round of representation collection and training shared across all in-domain datasets, while the baselines are run and trained/calibrated separately for each task, as they are originally designed, and their offline cost is obtained by averaging over the five ID tasks. From the results, our ICR, as a train-once-and-reuse method, has a comparable GPU hour cost to the per-task averages reported for the baselines (except I2CL, which only performs calibration rather than training).. This implies that once two or more tasks are evaluated, the amortized cost of ICR becomes lower, making our method more time-efficient in practice.

Table 10: GPU hours comparison across methods.

| Method | I2CL | LIVE | M2IV | ICR |
|---|---|---|---|---|
| GPU Hours | 1.2h | 6.8h | 7.0h | 9.7h |

Table 11: **Cached parameter size** of different methods. $M$ = #demonstration tokens, $d$ = hidden dimension, $L$ = #layers, $r$ = PID subspace rank ($r \ll M$).

| Method | Zero-shot | Few-shot | TV | FV | ICV | I2CL | LIVE | M$^2$IV | ICR |
|---|---|---|---|---|---|---|---|---|---|
| Cached Param. | 0 | $2MdL$ | $d$ | $d$ | $dL$ | $2dL$ | $dL$ | $2dL$ | $2rdL$ |

# G  ADDITIONAL ABLATION STUDY

## G.1  PIDs EXTRACTION

In Sec. 4.3 we reported the impact of varying the PCA rank and replacing PCA with a random basis. Here we provide additional details and observations.

For the random orthogonal subspace ($r = 8$), we generate a $d \times r$ Gaussian matrix per layer and apply QR decomposition to obtain an orthogonal basis. This ensures the comparison isolates the role of PCA-extracted directions from generic low-rank projections.

While Sec.4.3 reports the performance trade-offs, we note that the degradation at $r = 12$ is not only consistent across settings but also more unstable across runs, suggesting that the enlarged subspace introduces degrees of freedom that remain under-trained with fixed data and epochs. This further supports the interpretation that OOD robustness benefits from a carefully constrained subspace.

Although in-domain accuracy is relatively preserved under the random basis (indicating the model can adapt with enough supervision), both near- and far-OOD performance collapse. This highlights that OOD generalization is not a byproduct of low-rank routing alone: it specifically requires alignment with meaningful directions identified by PCA. Without such alignment, routing vectors fail to capture exemplar-derived cues, and the model effectively loses its cross-task transfer ability.

## G.2  ICD SAMPLING

We vary strategies for constructing ICL prompts in PIDs extraction. Specifically, BALANCE/$k$ denotes sampling $k$ ICDs per class in a balanced manner, while SIMILARITY selects ICDs based on BERT embedding similarity to the query (Liu et al., 2021), with the total number of ICDs matched to that of BALANCE/5. Table 12 shows that although SIMILARITY performs comparably in-domain, it substantially degrades near- and far-OOD accuracy, indicating overfitting to query-local patterns rather than capturing cross-domain invariances. This result highlights that exemplar diversity, rather than local similarity, is most critical for robust PIDs extraction. Within the balanced scheme, $k = 5$ achieves the best trade-off: fewer exemplars ($k = 3$) reduce coverage, while more ($k = 7$) add redundancy without benefit.

## G.3  ROUTING LAYERS

We investigate the effect of applying ICR at different depths within the model by evenly dividing it into early, middle, and late segments. Table 13 shows that intervening at the late layers yields the best overall performance. This outcome reflects a fundamental difference between ICR and prior vector-based methods like I2CL. Vector-based approaches add interventions on the residual stream whose effects tend to accumulate linearly, making adjustments from early or middle layers relatively stable. In contrast, ICR directly modulates Q/K alignment via gated subspace coefficients. The resulting changes to attention distributions are nonlinear and softmax-amplified, which may propagate through subsequent layers. When such routing is altered too early, small misalignments can cascade and erode the low-level syntactic structure, causing all settings that involve early-layer intervention (including Early and All) to collapse. Focusing the intervention on late layers instead acts as a high-level readout reweighting, preserving early representations while concentrating adaptation near semantic integration and decision formation.

## G.4  INFORMATION USAGE IN PID EXTRACTION

In PIDs extraction, we collect Q/K representation from the last token of ICL prompts. To explore alternative ways of extracting Q/K representations Specifically, we test (i) mean pooling over the last

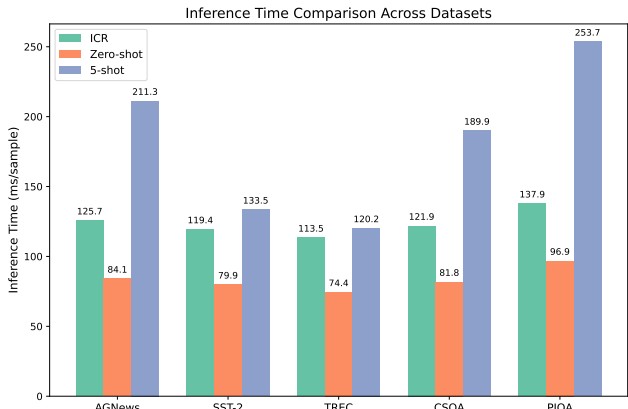

Figure 5: Comparison of average per-sample inference time across five datasets for 5-shot, zero-shot, and ICR methods.

Table 12: Ablation on ICD sampling in ICL bases construction. Scores are averaged over ID, near-OOD, and far-OOD groups.

| Method | ID | Near OOD | Far OOD |
|---|---|---|---|
| SIMILARITY | 67.3 | 55.0 | 49.4 |
| BALANCE/3 | 62.1 | 52.4 | 48.8 |
| BALANCE/5 | **67.7** | **57.3** | **52.0** |
| BALANCE/7 | 66.8 | 56.9 | 50.2 |

Table 13: Ablation on routing layers. Scores are averaged over ID, near-OOD, and far-OOD groups.

| Layers | ID | Near OOD | Far OOD |
|---|---|---|---|
| Early | 40.6 | 47.7 | 41.0 |
| Middle | 60.3 | 56.3 | 37.3 |
| Late | **67.7** | **57.3** | **52.0** |
| All | 48.6 | 41.4 | 40.2 |

4 tokens, (ii) mean pooling over the last 8 tokens, and (iii) an attention-rollout–based pooling that aggregates token-level Q/K using attention-flow weights computed across all layers. Conceptually, the rollout variant constructs a cumulative attention map by multiplying layer-wise attention matrices and uses the resulting contribution scores to weight each token's Q/K before pooling. Experimental results for the above variants are reported in Table 14.

Across all benchmarks including ID, near-OOD, and far-OOD, none of the alternative pooling strategies outperform the last-token extraction. Performance degrades as the pooling window expands, and the attention-rollout variant yields the weakest results. This pattern suggests that incorporating a broader set of tokens introduces noise from heterogeneous token roles, diluting the ICL-related signal that PIDs aim to isolate. A clear trend emerges that the more tokens included in the pooling region, the more the essential alignment signal is blurred. The effectiveness of the last-token extraction is actually consistent with the functional role of this position in ICL. The final token before answering is where the model synthesizes the full prefix (query and demonstrations) into a single attention computation immediately before prediction. This makes it a coherent integration point where the demonstration-induced structure is concentrated. Moreover, it is precisely the position at which ICR injects its attention-logits bias during inference. Extracting PIDs from the same locus where the intervention is later applied provides a natural alignment between the raw attention geometry and the added low-rank bias. These results indicate that the last-token Q/K captures the most stable and transferable ICL-related structure, while broader pooling mixes in context that is not directly relevant for the ICL computation. From an interpretability perspective, the fact that PIDs extracted at the same position where we intervene work best shows that ICR is indeed leveraging the attention structure that few-shot ICL forms at this locus.

## G.5 TEXT ENCODER

To assess whether the choice of frozen text encoder affects routing quality and cross-domain generalization, we conduct an ablation in which we replace `all-MiniLM-L6-v2` with a stronger encoder, `all-mpnet-base-v2`. The latter has more layers and a higher embedding dimension (768 vs. 384), providing richer semantic representations at the cost of slower encoding.

Table 14: Effect of different Q/K pooling strategies for PID extraction on ICR performance.

| Method | In-Domain (ID) | | | | | Near OOD | | | Far OOD | | | | Average |
|---|---|---|---|---|---|---|---|---|---|---|---|---|---|
| | AG | SST-2 | TREC | CSQA | PIQA | SST-5 | MR | MRPC | CB | COPA | CREAK | AI2SciE | |
| Default (last token) | 86.6 | 86.4 | 83.8 | 24.8 | 57.0 | 38.6 | 79.8 | 53.4 | 46.4 | 68.0 | 56.4 | 37.2 | 59.9 |
| Last 4 tokens | 84.6 | 83.2 | 83.8 | 25.6 | 55.0 | 35.6 | 73.8 | 46.6 | 39.3 | 61.0 | 53.4 | 31.2 | 56.1 |
| Last 8 tokens | 67.8 | 77.2 | 71.4 | 23.8 | 54.0 | 27.6 | 70.4 | 45.6 | 28.6 | 62.0 | 55.8 | 31.6 | 51.3 |
| Attention rollout | 67.2 | 78.8 | 67.0 | 22.2 | 52.6 | 26.8 | 72.8 | 44.4 | 32.2 | 62.0 | 52.4 | 32.8 | 50.9 |

Table 15: Effect of frozen text encoder choice on ICR performance.

| Encoder | In-Domain (ID) | | | | | Near OOD | | | Far OOD | | | | Average |
|---|---|---|---|---|---|---|---|---|---|---|---|---|---|
| | AG | SST-2 | TREC | CSQA | PIQA | SST-5 | MR | MRPC | CB | COPA | CREAK | AI2SciE | |
| miniLM | 86.6 | 86.4 | 83.8 | 24.8 | 57.0 | 38.6 | 79.8 | 53.4 | 46.4 | 68.0 | 56.4 | 37.2 | 59.9 |
| mpnet | 86.8 | 87.0 | 88.2 | 25.0 | 58.6 | 37.0 | 86.4 | 56.6 | 48.2 | 64.0 | 57.0 | 38.0 | 61.1 |
| Δ | +0.2 | +0.6 | +4.4 | +0.2 | +1.6 | -1.6 | +6.6 | +3.2 | +1.8 | -4.0 | +0.6 | +0.8 | +1.2 |

The results, summarized in Table 15, show that `all-mpnet-base-v2` yields slightly better performance in both ID and OOD settings, while the overall trend and relative performance of ICR remain consistent. This indicates that (i) the router is able to effectively exploit the semantic features provided by the frozen encoder, and (ii) ICR's generalization behavior is robust to the encoder choice rather than being tied to a particular model. As expected, larger encoders offer marginally better semantic retrieval at the cost of slower usage, so the choice involves a tradeoff between speed and capacity.

## H "ICLNESS" TOKENS

For each dataset $d$ (including all ID, near-OOD, and far-OOD tasks), we run the model in both zero-shot and ICR-augmented settings, compute the next-token log-probabilities, and obtain

$$\Delta \log p^{(d)} = \log p_{\text{ICR}} - \log p_{\text{zs}}.$$

Averaging over all examples in $d$ yields a token-level bias vector $b^{(d)} \in \mathbb{R}^{|\mathcal{V}|}$, where each coordinate indicates the systematic up- or down-weighting of a token by ICR on that dataset. We then aggregate across datasets with the following statistics for each vocabulary token $v$:

- $\text{mean}_v$: mean $\Delta \log p$ across datasets

- $\text{std}_v$: standard deviation across datasets

- $\text{pos\_rate}_v$: fraction of datasets with $\Delta \log p > 0$

- $\text{borda}_v$: Borda rank fusion across datasets

- $\text{stability}_v = \text{mean}_v / (\text{std}_v + \epsilon)$

The final score is defined as

$$\text{score}_v = \text{stability}_v \cdot \text{pos\_rate}_v \cdot \log(1 + \text{borda}_v),$$

which rewards tokens that are (i) strongly upweighted on average, (ii) consistently positive across datasets, and (iii) highly ranked across tasks. The top-50 tokens are listed in Table 16, with tokens strongly related to in-context reasoning or structural semantics ("ICLness tokens") highlighted in red.

One might argue that because we explicitly require consistency across datasets, the resulting tokens are trivially "cross-dataset". However, cross-dataset consistency alone does not guarantee interpretability: many tokens that satisfy this criterion are function words (e.g., the, and) or generic terms (e.g., *people*, *year*) that carry little connection to in-context reasoning. The notable observation is that the tokens emerging at the very top of the ranking are not such trivial items, but words with structural and explanatory semantics (e.g., *illustrated*, *constitution*, *protected*). This indicates that ICR does not merely enforce consistency on generic vocabulary, but systematically biases the model toward dimensions plausibly linked to reasoning and explanation, aligning with our hypothesis about generalizable "ICLness."

Table 16: Top-50 dataset-invariant "ICLness" tokens. A higher score indicates a more stable and consistent positive bias across ID, near-OOD, and far-OOD datasets.

| Rank | Token | Score | Mean $\Delta \log p$ | Std | Pos. Rate | Borda Norm |
|------|-------|-------|------|-----|-----------|------------|
| 1 | dep | +28.79 | +0.73 | 0.02 | 1.00 | 0.825 |
| 2 | court | +22.31 | +0.75 | 0.02 | 1.00 | 0.828 |
| 3 | *forme* (French form) | +21.92 | +0.74 | 0.02 | 1.00 | 0.823 |
| 4 | *illustrated* | +19.80 | +0.21 | 0.00 | 1.00 | 0.538 |
| 5 | *constitution* | +18.92 | +0.48 | 0.01 | 1.00 | 0.704 |
| 6 | *protected* | +18.35 | +0.75 | 0.02 | 1.00 | 0.829 |
| 7 | network | +17.01 | +0.76 | 0.03 | 1.00 | 0.836 |
| 8 | thoughts | +13.51 | +0.47 | 0.02 | 1.00 | 0.695 |
| 9 | colonial | +13.49 | +0.71 | 0.03 | 1.00 | 0.815 |
| 10 | drie | +13.41 | +0.72 | 0.03 | 1.00 | 0.816 |
| 11 | acres | +12.50 | +0.50 | 0.02 | 1.00 | 0.711 |
| 12 | fro | +12.22 | +1.11 | 0.06 | 1.00 | 0.934 |
| 13 | *protection* | +12.14 | +0.83 | 0.04 | 1.00 | 0.861 |
| 14 | reve | +11.79 | +0.68 | 0.03 | 1.00 | 0.797 |
| 15 | leur | +11.14 | +0.70 | 0.04 | 1.00 | 0.809 |
| 16 | *trouv* (French find) | +10.72 | +0.77 | 0.04 | 1.00 | 0.839 |
| 17 | *clause* | +10.09 | +0.56 | 0.03 | 1.00 | 0.744 |
| 18 | pipe | +10.07 | +1.12 | 0.07 | 1.00 | 0.923 |
| 19 | *column* | +10.04 | +0.52 | 0.03 | 1.00 | 0.723 |
| 20 | Tot | +9.21 | +0.33 | 0.01 | 1.00 | 0.618 |
| 21 | catt | +9.17 | +1.01 | 0.07 | 1.00 | 0.914 |
| 22 | networks | +9.16 | +0.69 | 0.04 | 1.00 | 0.805 |
| 23 | cyl | +9.12 | +1.28 | 0.09 | 1.00 | 0.958 |
| 24 | duch | +8.69 | +0.87 | 0.06 | 1.00 | 0.868 |
| 25 | bro | +8.67 | +0.32 | 0.02 | 1.00 | 0.609 |
| 26 | *enumerate* | +8.54 | +0.45 | 0.03 | 1.00 | 0.686 |
| 27 | surv | +8.34 | +0.74 | 0.05 | 1.00 | 0.824 |
| 28 | burst | +8.27 | +0.65 | 0.05 | 1.00 | 0.788 |
| 29 | *connections* | +8.08 | +0.85 | 0.07 | 1.00 | 0.868 |
| 30 | *presente* (French present) | +8.08 | +0.59 | 0.04 | 1.00 | 0.760 |
| 31 | colors | +7.99 | +0.63 | 0.05 | 1.00 | 0.776 |
| 32 | *signs* | +7.78 | +0.41 | 0.03 | 1.00 | 0.662 |
| 33 | *filter* | +7.55 | +1.07 | 0.09 | 1.00 | 0.916 |
| 34 | indust | +7.37 | +0.26 | 0.02 | 1.00 | 0.571 |
| 35 | *returns* | +7.24 | +0.88 | 0.08 | 1.00 | 0.879 |
| 36 | *filters* | +7.23 | +1.19 | 0.11 | 1.00 | 0.943 |
| 37 | alles | +7.22 | +0.88 | 0.08 | 1.00 | 0.880 |
| 38 | *zusammen* (German jointly) | +7.11 | +0.74 | 0.06 | 1.00 | 0.820 |
| 39 | neces | +7.08 | +0.94 | 0.08 | 1.00 | 0.886 |
| 40 | tandis | +7.07 | +0.85 | 0.08 | 1.00 | 0.867 |
| 41 | *separately* | +6.94 | +1.14 | 0.11 | 1.00 | 0.946 |
| 42 | bird | +6.69 | +0.42 | 0.03 | 1.00 | 0.670 |
| 43 | blieb | +6.57 | +0.52 | 0.04 | 1.00 | 0.722 |
| 44 | *comprend* (French comprehend) | +6.53 | +0.93 | 0.09 | 1.00 | 0.888 |
| 45 | *contrib* | +6.45 | +0.60 | 0.05 | 1.00 | 0.765 |
| 46 | *capture* | +6.41 | +0.57 | 0.05 | 1.00 | 0.745 |
| 47 | strict | +6.40 | +0.73 | 0.07 | 1.00 | 0.813 |
| 48 | happy | +6.28 | +0.45 | 0.04 | 1.00 | 0.681 |
| 49 | lange | +6.21 | +0.55 | 0.05 | 1.00 | 0.744 |
| 50 | condem | +6.18 | +0.64 | 0.06 | 1.00 | 0.789 |

## I LAYER IMPORTANCE

Figures 6a and 6b report the normalized layer-importance profiles across all in-domain (ID) and out-of-domain (OOD) datasets, respectively. Each curve corresponds to one dataset, and the $x$-axis denotes the global transformer layer index. By comparing the two figures, several observations can be made. First, both ID and OOD datasets consistently highlight a few dominant "hub" layers (e.g., around layers 23 and 26), indicating that ICR relies on these shared layers as primary routing points. Notably, such hub layers are concentrated in the earlier–middle part of the intervened layers, while later layers no longer exhibit clear global hubs, suggesting that they play a more task-specific role.

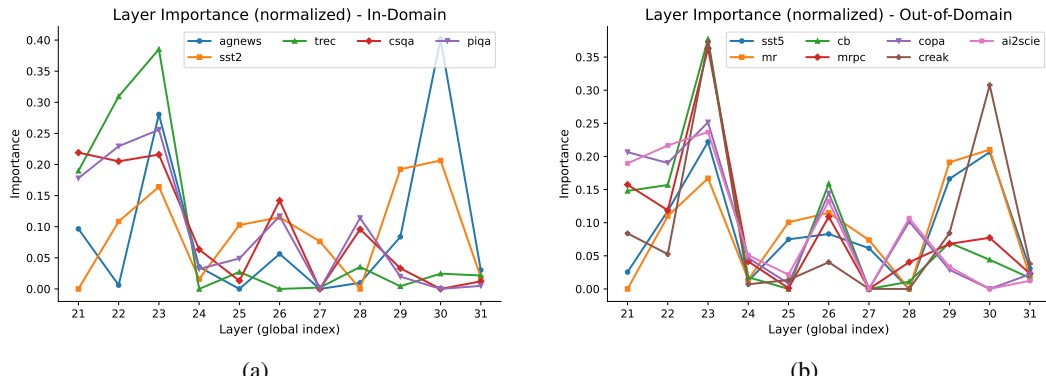

(a)  (b)

Figure 6: Layer importance profiles. Curves show per-layer importance, computed from head gates and routing coefficients.

Second, certain OOD datasets exhibit importance profiles that closely resemble those of particular ID datasets, suggesting that ICR is able to adjust its routing behavior in a task-aware manner rather than collapsing to a uniform pattern. Third, the importance peaks in OOD settings are sharper, implying that under distribution shift the model leans more heavily on these hub layers as stable anchors to preserve generalization.

## J  ADDITIONAL RELATED WORK

**Mechanisms of In-context Learning.**  To better exploit ICL, considerable efforts have been devoted to understanding the mechanisms of ICL (Li et al., 2025c;b). ICL was initially regarded as an ability that emerges as LLMs scale up in parameters and training data (Wei et al., 2022). Subsequent work has sought to provide theoretical interpretations through two main perspectives. Garg et al. (2022) modeled ICL as a form of gradient descent. Based on this, Von Oswald et al. (2023); Dai et al. (2022) explained ICL via meta-optimization. Alternatively, Xie et al. (2021) framed ICL as implicit Bayesian inference, suggesting that LLMs infer a shared latent concept across ICDs. Beyond modeling of model behavior, the connection between MHA and ICL has also been extensively studied. Induction heads, which are attention heads that learn repeated patterns in the prompt and are considered key contributors to ICL, were identified by Elhage et al. (2021) and empirically analyzed by Olsson et al. (2022). Todd et al. (2023) further employed causal mediation analysis to identify the heads that contribute most to ICL, denoted as FV heads. Yin & Steinhardt (2025) provided a systematic synthesis of these findings. In contrast to these works, we develop a deeper theoretical framework for ICL through attention routing, which can be effectively applied to enhance ICL performance. Whether ICL can truly generalize to OOD tasks is another central question. Yadlowsky et al. (2023) find that ICL struggles to generalize to function classes unseen during training, such as convex combinations or extreme variants of the pretraining functions. Wang et al. (2024b) further argue that ICL fails to generalize to new task instances even within a seen distribution, instead exposing its limitation in handling unseen input–label distributions.

## K  MECHANISM DISCUSSION

One might argue that certain attention behaviors in transformers, such as induction heads, implement global attention from demonstrations to the query, and that a method like ICR, which operates under zero-shot inputs without explicit demonstrations, should be unable to reconstruct such patterns. Yet empirically, ICR can match or even surpass vanilla few-shot ICL in several settings, which calls for a more refined view of what demonstrations contribute.  Our position is that, in the zero-shot regime, the benefits commonly attributed to demonstrations in vanilla ICL can be reinterpreted as local, intra-query attention routing. ICR explicitly operates in this regime that it does not reconstruct demo-to-query links, instead, it modulates attention logits within the query so that the model allocates attention along task-useful paths. Concretely, the low-rank update $\Delta A$ encodes cross-task, reusable priors over intra-query routing, learned from pooled Q/K statistics across tasks, such as: **(i) role-typing** (e.g., anchors such as question stems, label markers, options, premises vs. hypotheses); **(ii) long-range links between these roles** (e.g., question↔option, premise↔hypothesis, number↔unit); and **(iii) competition/sparsity priors** that sharpen relevant links and suppress distractors. Applying $\Delta A$ rotates the query–key geometry to reinstate these priors on a new query,

yielding attention maps that functionally resemble those induced by good demonstrations, without requiring any demo content. This explains why zero-shot ICR can match or exceed vanilla few-shot when demonstrations are noisy or misaligned. Importantly, $\Delta A$ transfers a routing prior rather than learning new content at inference time. When a task truly relies on demo-specific content (beyond routing), explicit few-shot prompting can be stronger, since such content cannot be reinstated by intra-query attention routing alone. This explains why, on in-domain benchmarks, ICR may underperform vanilla few-shot ICL: some queries benefit directly from the information contained in the demonstrations. By contrast, in OOD settings the main benefit of few-shot prompting often lies in inducing a robust intra-query routing pattern, while its demo content can be misaligned or even misleading. By extracting and reusing this pattern, ICR attains few-shot–like gains without exposure to OOD demo-content mismatch, yielding more comparable and stable performance under distribution shift. This may provide a useful new perspective for future work on understanding the mechanisms underlying in-context learning.

## THE USE OF LARGE LANGUAGE MODELS (LLMS)

In preparing this submission, we used large language models (LLMs) solely for language refinement. Specifically, LLMs were employed to polish the writing style and improve readability, such as rephrasing sentences and adjusting grammar.

