# OpenReview forum: "Towards Generalizable Implicit In-Context Learning with Attention Routing"
_ICLR.cc/2026/Conference — Submitted to ICLR 2026_

### Official Review · Reviewer_LNqB · 2025-10-16

**Soundness:** 3
**Presentation:** 2
**Contribution:** 3
**Rating:** 6
**Confidence:** 4

**Summary:**

This paper focuses on reconstructing the attention patterns of few-shot inputs on zero-shot settings. Specifically, this paper propose In-context Routing method, which adds a bias term to the attention logits generated in zero-shot inputs to reconstruct the attention patterns observed in few-shot scenarios. Extensive quantitative experiments demonstrate that the proposed method outperforms a wide range of modern baselines, especially on out-of-domain tasks.

**Strengths:**

1. The authors propose a novel framework called attention routing, which enables automatic additive steering of attention logits, which can be broadly applicable across various scenarios. The attempt to explicitly control LLMs' behavior through a mechanistic understanding is highly revolutionary for the field of interpretability. This is my primary reason for recommending the acceptance of this paper.

2. Based on the above attention routing framework, the authors further introduce the In-context Routing (ICR) method. Through extensive quantitative experiments on sufficiently diverse datasets and model types, the authors demonstrate that ICR outperforms multiple baselines.

3. The analysis section provides insightful details about the proposed method, strengthening their claim that ICR provides generalizable attention shaping. In particular, they find that the reshaped attention scores can capture reasoning-oriented tokens, thereby confirming the soundness of the original motivation to ICR.

**Weaknesses:**

1. ICR utilizes gradient-based training on relatively large datasets with several tricks to facilitate attention routing. Also, ICR introduces an external text encoder to calculate the two key control gates in the method, and optimizes on a complex loss function. This design may contradict the low-resource spirit of ICL and undermine the overall usability. Furthermore, to my knowledge, the authors did not discuss how the performance of this additional text encoder affects ICR, nor did they provide sufficiently convincing ablation results to confirm the effectiveness of each loss component (e.g., in line 2-4 of Table 4, ablating some loss terms does not harm the accuracy). I consider attention routing to be an elegant framework, but relying on a bulky auxiliary module seems less than ideal.

    At the same time, this raises concerns regarding the paper’s main results (Tables 1 and 2): many of the provided baselines (such as TV and FV) involve substantially lower computational costs than ICR, making the comparison somewhat unfair. Although the authors claim that ICR exhibits good generalization and reusability, I would like to see at least a comparison in terms of calculation cost to enhance the credibility of their results.

    Moreover, from another perspective, since ICR already uses gradient-based training, it can be reasonable to directly train $\Delta \mathbf{A}$. I hope the authors can include such an experiment to demonstrate that their manual selection of the $\Delta \mathbf{A}$ basis is not redundant.

2. Mechanistically, the authors employ an external text encoder ($E(\cdot)$) to predict two key gating units within the ICR framework. These gating units are closely related to the internal structure of the LLM (e.g., selecting the important heads, as the authors mentioned in Section 5.3). Therefore, a crucial question arises: do the $E(x)$ actually contain information about the LLM’s internal structure? Or is $E(x)$ merely irrelevant variables? A simple experiment could address this by ablating $E(x)$ into random vectors. If the former is the case, how is this information then extracted by the two parameters $\theta$? This should be an interesting analysis, yet the authors skipped it.

3. There are several writing issues that make the paper somewhat difficult to follow, but I believe that this does not significantly affect my overall judgment of the paper.

    1. Line 52. “out-of-domain (OOD)” is ambiguous. You seem to mean that the query lies outside the distribution of the demonstrations, but another possible interpretation is that “the query lies outside the pre-training distribution”. Understanding this is crucial to getting your motivation, so I recommend clarifying it to eliminate ambiguity. Also, the specific experimental setup of Fig. 1 should be described (perhaps in the appendix).

    2. Line 115. This paragraph is somewhat unclear. I don’t fully understand the causal link in “Such additive interventions cannot structurally control how information flows, and thus often remain tied to task-specific representations.” I can understand that using task vectors for steering cannot *explicitly* control the information flow (i.e., attention scores, although not absolutely, since injecting certain components into the attention query could indirectly alter the attention scores), but I don’t see how this leads to being “tied to task-specific representations.” If I have missed something, I apologize.

    3. I suggest that the authors explain how each introduced symbol is grounded. For example, the symbol $\alpha$ in Equation (3) is confusing, it isn’t clear until Sec. 3.2 introduce it as a parameter to be trained.

**Questions:**

1. The authors seem to attribute all the benefits of ICL demonstrations to local attention effects within the query’s tokens (i.e., dynamically filtering task-relevant signals through attention scores). However, as far as I know, additional attention behaviors such as induction heads perform global attention operations from the demonstrations to the query. ICR clearly cannot reconstruct such attention patterns, since it is conducted under zero-shot inputs, yet their method still outperforms vanilla few-shot. This might prompt a new perspective on the mechanism of ICL. I would like to ask how the authors interpret this phenomenon, and whether they could expand their discussion of such mechanisms in the paper.

2. The analysis of layer/head importance (Fig. 4, left and middle) appears to include only the later layers. Could you release the results for all layers? This seems to suggest that certain specific attention heads induce the 0-shot inference, which can thus be improved by attention routing, therefore, it is interesting to get the detailed distribution of such heads.

---

> ### Author Response · Authors · 2025-11-21
>
> We thank the reviewer for the insightful and valuable comments. We respond to each comment as follows and sincerely hope that our rebuttal properly addresses your concerns. If so, we would deeply appreciate it if you could raise your score. If not, please let us know your further concerns, and we will continue actively responding to your comments and improving our submission.
>
> **Weakness1 (Part1)**
> > ICR utilizes gradient-based training on relatively large datasets with several tricks to facilitate attention routing. Also ICR introduces an external text encoder to calculate the two key control gates in the method, and optimized on a complex loss function. This design may contradict the low-resource spirit of ICL and undermines the overall usability.
>
> **R1:**
> Thank you for the comment. We agree that ICR introduces a lightweight text-encoder–based router trained offline. While this adds training, **the only trainable part is a small text encoder, and it removes the need to maintain a demo database.**
>
> Our focus is low resource at inference that once the router is learned, ICR requires no retrieval and no long demonstration context at test time, which reduces latency and memory pressure compared to explicit ICL. Actually, training-in-the-loop is **common** in implicit-ICL work (e.g., M2IV [1], LIVE[2]). Relative to those, ICR trains once and then generalizes across multiple ID/OOD benchmarks.
>
> Another goal is to broaden ICL’s benefit from “generalize within a task” to “generalize **from a few tasks to many task categories**.” A straightforward baseline is explicit cross-task ICL, but our preliminary results (Fig. 1) show it is sensitive to demonstration noise and fails to consistently transfer. This motivates extracting a reusable ICL pattern and training a small router to apply it in an input-conditioned way.
>
> References:
>
> [1] Yanshu Li, Yi Cao, Hongyang He, et al. M2IV: Towards efficient and fine-grained multimodal in-context learning via representation engineering. In Second Conference on Language Modeling, 2025
>
> [2] Yingzhe Peng, Xinting Hu, Jiawei Peng, Xin Geng, Xu Yang, et al. Live: Learnable in-context vector for visual question answering. Advances in Neural Information Processing Systems, 37: 9773–9800, 2024.
>
> **Weakness1 (Part2)**
> > Furthermore, to my knowledge, the authors did not discuss how the performance of this additional text encoder affects ICR, nor did they provide sufficiently convincing ablation results to confirm the effectiveness of each loss component (e.g., in the line 2-4 of Table 4,ablating some loss terms does not harm the accuracy). I consider attention routing to be an elegant framework, but relying on a bulky auxiliary module seems less than ideal.
>
> **R2:**
> We thank the reviewer for this insightful comment. **Regarding the first point**, we explicitly examined how the text encoder impacts ICR by swapping the default lightweight MiniLM-L6 (384d) with a stronger MPNet-base (768d) under identical settings, using Llama2-7B as backbone. The stronger encoder yields consistently better accuracy on most benchmarks, indicating that the semantic information extracted by the encoder is indeed exploited by ICR to choose different PID combinations and head-gating strategies conditional on the query. However, the gains are modest overall. For efficiency and a better accuracy–latency trade-off, we retain the lightweight MiniLM-v2 as our main configuration while acknowledging that larger encoders provide incremental headroom. We have added this into Appendix G.5 in the revision.
>
> | encoder\task      | agnews | sst2 | trec | csqa | piqa | sst5 | mr   | mrpc | cb   | copa | creak | ai2scie | Avg  |
> |-----------|-------:|-----:|-----:|-----:|-----:|-----:|-----:|-----:|-----:|-----:|------:|--------:|-----:|
> | all-MiniLM-L6-v2      | 86.6 | 86.4 | 83.8 | 24.8 | 57.0 | 38.6 | 79.8 | 53.4 | 46.4 | 68.0 | 56.4  | 37.2    | 59.9 |
> | all-mpnet-base-v2     | 86.8 | 87.0 | 88.2 | 25.0 | 58.6 | 37.0 | 86.4 | 56.6 | 48.2 | 64.0 | 57.0  | 38.0    | 61.1 |
> | Δ                   | +0.2 | +0.6 | +4.4 | +0.2 | +1.6 | -1.6 | +6.6 | +3.2 | +1.8 | -4.0 | +0.6  | +0.8    | +1.2 |
>
>
> **Regarding the second point**, we would like to clarify that the statement “ablating some loss terms does not harm accuracy” **only holds for in-domain settings**. As analyzed in paragraph "Key Components" in Section 4.3, removing Lconf, Lspar, or Lgate results in consistent degradation on OOD, especially far-OOD, benchmarks. These auxiliary losses are designed to **prevent shortcut learning** by producing over-uncertain predictions that lower entropy or by overfitting through excessive editing, and to enhance cross-domain generalization. Their ablation consistently harms OOD transfer, validating their intended effect.

---

> ### Author Response · Authors · 2025-11-21
>
> **Weakness1 (Part3)**
> > Although the authors claim that ICR exhibits good generalization and reusability, I would like to see at least a comparison in terms of calculation cost to enhance the credibility of their results.
>
> **R3：**
> We thank the reviewer for raising this important point about computational cost. First, ICR follows a train-once-and-reuse paradigm: unlike dataset-specific retrieval or per-task training baselines, its training cost is incurred **only once**, and the resulting router can be reused across all datasets. Second, training components are also common in recent implicit ICL methods. Strong baselines such as M2IV and LIVE are likewise training-based and therefore have computational costs comparable to ICR. Furthermore, we included in Table 9 (Appendix F) a comparison of cached parameter size, showing that ICR is in the same order of magnitude as implicit-ICL baselines (I2CL, LIVE, M2IV).
>
> To address the reviewer’s concern more concretely, we provide an explicit computational cost comparison with I2CL, M2IV, and LIVE. Since all these baselines and ICR are implicit ICL methods that perform zero-shot inference at test time with only a very small runtime overhead relative to standard zero-shot inference (see Fig.5), we focus on the **offline computational cost**, measured in NVIDIA V-100 GPU hours for representation collection and training/calibration. Because ICR extracts PIDs and trains the router using five in-domain datasets, and the baselines are not inherently designed for OOD scenarios, we report their offline cost on ID tasks only. Concretely, ICR performs a single round of representation collection and training shared across all in-domain datasets, while the baselines are run and trained/calibrated separately for each task, as they are originally designed, and their offline cost is obtained by averaging over the five ID tasks. The LLM used is Llama2-7B.
>
> From the results, our ICR, as a train-once-and-reuse method, has a comparable GPU hour cost to the **per-task averages** reported for the baselines (except I2CL, which only performs calibration rather than training). This implies that once two or more tasks are evaluated, the amortized cost of ICR becomes lower, making our method more **time-efficient** in practice.
>
> We have added this to Appendix F in the revision.
>
> |   Method        | I2CL    |  LIVE   |  M2IV   |   ICR   |
> | --------- | --- | --- | --- | ---- |
> | GPU Hours | 1.2h   |  6.8h   |  7.0h   | 9.7h |
>
> **Weakness1 (Part4)**
> > It can be reasonable to directly train ΔA. I hope the authors can include such an experiment to demonstrate that their manual selection of the ΔA basis is not redundant.
>
> **R4：**
> We thank the reviewer for this insightful comment regarding the role of extracted PIDs as the basis of ΔA.
> We already include an ablation in Section 4.3 that replaces the PCA-selected basis with a **random orthogonal basis**. Under the same rank and training setup, this variant shows clear performance drops across in-domain, near-OOD, and far-OOD benchmarks (with smaller degradation on ID). This suggests that gradient-based training can partially adapt to a random basis under supervision, but the learned ΔA fails to transfer effectively to OOD tasks, confirming the necessity of a PCA-based basis.
>
> Following the suggestion, we implemented a fully trainable relative-position bias (ΔA) after each attention head in the last one-third of layers, trained jointly on the five ID datasets under the same epochs as ICR. Each selected layer–head learns an additive bias over relative offsets in attention logits with a tanh-squashed parameterization, L1 regularization on the bias, and the same cross-entropy plus confidence loss as ICR. This variant increases parameters by (layers in last third) × (heads) × (2·max_len−1), entailing a strictly higher computational cost than ICR’s low-rank routing.
>
> | Settings                | In-domain | Near-OOD | Far-OOD |
> |-------------------------|----------:|---------:|--------:|
> | ICR default             | **67.7**      | **57.3**     | **52.0**    |
> | Rand. Orth.             | 63.9      | 48.1     | 46.7    |
> | Fully trainable ΔA      | 64.7      | 45.7     | 42.1    |
>
> Although fully trainable ΔA slightly improves ID over Random Orth., it falls behind on near- and far-OOD and remains consistently below ICR. This suggests that low-rank routing aligned with PCA-extracted PIDs supplies a crucial transfer-friendly inductive bias and query-conditional adaptation that fully trainable ΔA lacks. Moreover, with substantially more degrees of freedom, ΔA is prone to overfitting domain-specific position patterns, while under the same training budget it is also harder to fully calibrate. Hence the data-driven choice of the ΔA basis is not redundant. It is precisely what enables OOD robustness, whereas fully training ΔA does not compensate for the missing alignment and conditioning even at higher cost.

---

> ### Author Response · Authors · 2025-11-21
>
> **Weakness2**
> > Mechanistically, the authors employ an external text encoder (E(·)) to predict two key gating units within the ICR framework. These gating units are closely related to the internal structure of the LLM (e.g., selecting the important heads, as the authors mentioned in Section 5.3). Therefore, a crucial question arises: do the E(X) actually contain information about the LLM’s internal structure? Or is E(X) merely irrelevant variables? A simple experiment could address this by ablating E(X) into random vectors. If the former is the case, how is this information then extracted by the two parameters theta? This should be an interesting analysis, yet the authors skipped it.
>
> **R5：**
> We thank the reviewer for this insightful comment regarding the role of the external text encoder E(x). We want to clarify that E(x) does not encode the internal structure of the LLM, instead, it provides a task- and input-dependent representation that conditions the router’s behavior. During training, the router learns to use this representation to select and combine PIDs and attention heads that are most relevant for the current query, which is consistent with prior observations that certain attention heads are task-related in ICL.
>
> To further test whether E(x) provides meaningful conditioning or is merely an irrelevant variable, we re-trained and evaluated ICR with a **random encoder** that replaces the sentence embedding by per-sample, seed-deterministic random vectors (same dimensionality and training losses; evaluation uses the same random mapping). Compared to the semantic text-encoder setting, the random encoder substantially degrades performance on most tasks, often falling back toward the zero-shot baseline. These results show that E(x) is not an irrelevant variable. The semantic encoder supplies informative conditioning that ICR leverages to choose different PID combinations and head-gating strategies per query. The ablation we ran for Weakness 1, where a stronger text encoder yields slightly better ICR performance, also supports this conclusion.
>
> | task           | agnews | sst2 | trec | csqa | piqa | sst5 | mr   | mrpc | cb   | copa | creak | ai2scie |
> |----------------|-------:|-----:|-----:|-----:|-----:|-----:|-----:|-----:|-----:|-----:|------:|--------:|
> | zero-shot      | 67.0   | 78.6 | 56.6 | 22.4 | 52.2 | 25.8 | 72.2 | 44.4 | 37.5 | 63.0 | 51.8  | 34.8    |
> | text-encoder   | 86.6   | 86.4 | 83.8 | 24.8 | 57.0 | 38.6 | 79.8 | 53.4 | 46.4 | 68.0 | 56.4  | 37.2    |
> | Random encoder | 64.2   | 82.8 | 44.8 | 22.8 | 50.4 | 29.6 | 76.6 | 52.8 | 23.2 | 64.0 | 51.4  | 36.6    |

---

> ### Author Response · Authors · 2025-11-21
>
> **Weakness3**
> > There are several writing issues that make the paper somewhat difficult to follow, but I believe that this does not significantly affect my overall judgment of the paper.
>
> **R6:**
> We appreciate the reviewer’s comments on the writing clarity. We address these concerns as below.
>
> > Line 52. “out-of-domain (OOD)” is ambiguous. You seem to mean that the query lies outside the distribution of the demonstrations, but another possible interpretation is that “the query lies outside the pre-training distribution”. Understanding this is crucial to getting your motivation, so I recommend clarifying it to eliminate ambiguity. Also, the specific experimental setup of Fig. 1 should be described (perhaps in the appendix).
>
> In our paper, out-of-distribution (OOD) refers to tasks that fall outside the distribution of task categories used for **both PID extraction and pretraining**. Since the same set of tasks was used in both stages (see Sec. 5.2), we did not explicitly distinguish between the two distributions.
>
> Preliminary Experiment Setup:
> The inference-time model is Llama-2-7B, and the prompt template follows Table 6 in Appendix D.3.1. Each run uses a total of 16 shots. In the single-source setting, we sample 16 demonstrations without replacement from the source task’s train split; in the cross-task setting, we sample 8 demonstrations from each of two source tasks, concatenate them, and uniformly shuffle their order. In both settings, demonstrations from any given dataset are sampled with label-balanced sampling. For evaluation, each target is tested on a subset of 500 test instances, and we report accuracy averaged over 5 independent seeds. Decoding is greedy. We have added this into Appendix D.3.2 in the revision.
>
>
> > Line 115. This paragraph is somewhat unclear. I don’t fully understand the causal link in “Such additive interventions cannot structurally control how information flows, and thus often remain tied to task-specific representations.” I can understand that using task vectors for steering cannot explicitly control the information flow (i.e., attention scores, although not absolutely, since injecting certain components into the attention query could indirectly alter the attention scores), but I don’t see how this leads to being “tied to task-specific representations.” If I have missed something, I apologize.
>
> The additive interventions in Eq.2 operate on the residual stream after the attention operation has completed, so they do not directly modify the attention weights within that layer. As a result, the model continues to aggregate information according to the original attention routes, and the injected shift vector primarily biases the resulting representations toward a direction derived from a specific task. Although such residual shifts can indirectly influence downstream attention in later layers, they do not provide an explicit mechanism for reshaping token-to-token routing or for dynamically adapting head usage in response to new tasks or distributions. Consequently, the learned effect tends to remain tied to the task-specific representation geometry on which the vector was estimated, with limited capacity for structured, query-dependent adaptation under OOD shifts.
>
> > I suggest that the authors explain how each introduced symbol is grounded. For example, the symbol in Equation (3) is confusing, it isn’t clear until Sec. 3.2 introduce it as a parameter to be trained.
>
> The parameter α in Eq. 3 is introduced right before the equation (line 145) as the routing vector assigning weights to the PIDs at layer l.

---

> ### Author Response · Authors · 2025-11-21
>
> **Question1**
> > The authors seem to attribute all the benefits of ICL demonstrations to local attention effects within the query’s tokens (i.e., dynamically filtering task-relevant signals through attention scores). However, as far as I know, additional attention behaviors such as induction heads perform global attention operations from the demonstrations to the query. ICR clearly cannot reconstruct such attention patterns, since it is conducted under zero-shot inputs, yet their method still outperforms vanilla few-shot. This might prompt a new perspective on the mechanism of ICL. I would like to ask how the authors interpret this phenomenon, and whether they could expand their discussion of such mechanisms in the paper.
>
> **R7:**
> We thank the reviewer for the thoughtful question. Our position is that, in the zero-shot regime, the benefits commonly attributed to demonstrations in vanilla ICL can be reinterpreted as local, intra-query attention routing. ICR explicitly operates in this regime: **it does not reconstruct demo→query links, instead, it modulates attention logits within the query so that the model allocates attention along task-useful paths**.
>
> Concretely, the low-rank update ΔA encodes cross-task, reusable priors over intra-query routing, learned from pooled Q/K statistics across tasks, such as:
> (i) **role-typing** (e.g., anchors such as question stems, label markers, options, premises vs. hypotheses);
> (ii) **long-range link** between these roles (e.g., question-option, premise-hypothesis, number-unit);
> (iii) **competition/sparsity priors** that sharpen relevant links and suppress distractors.
>
> Applying ΔA rotates the query–key geometry to reinstate these priors on a new query, yielding attention maps that functionally **resemble those induced by good demonstrations** without requiring any demo content. This explains why zero-shot ICR can match or exceed vanilla few-shot when demonstrations are noisy or misaligned.
>
> ΔA transfers a routing prior rather than learning new content at inference. When a task truly relies on demo-specific content (beyond routing), explicit few-shot can be stronger, since such content cannot be reinstated by intra-query attention routing alone. This explains why, on in-domain benchmarks, ICR may underperform vanilla few-shot: some queries benefit directly from the information contained in demonstrations.
>
> By contrast, in OOD settings the main benefit of few-shot often lies in inducing a robust intra-query routing pattern, while its demo content can be misaligned and even misleading. By extracting and reusing this pattern, ICR attains few-shot–like gains without exposure to OOD demo-content mismatch, yielding more comparable performance. We have added this mechanistic discussion in Appendix K in the revision.
>
> **Question2**
> > The analysis of layer/head importance (Fig. 4, left and middle) appears to include only the later layers. Could you release the results for all layers? This seems to suggest that certain specific attention heads induce the 0-shot inference, which can thus be improved by attention routing, therefore, it is interesting to get the detailed distribution of such heads.
>
> **R8:**
> We thank the reviewer for the insightful comment that our visualization suggests that certain attention heads play a critical role in enabling zero-shot inference, which can thus be enhanced through attention routing. This is precisely what we aim to illustrate. However, as described in Sec. 4.1 (Training) and Appendix G.3 (Table 11), ICR applies interventions only to the last one-third of layers, where we observed the strongest influence on performance. Therefore, the visualization in Fig. 4 focuses on this subset to highlight the layers and heads that are actually modulated by ICR.

---

> > ### Comment · Reviewer_LNqB · 2025-11-24
> >
> > Thanks for your reply. I think the authors have addressed most of my concerns. Therefore, I will raise the score to 8.

---

> > > ### Author Response · Authors · 2025-11-24
> > >
> > > Thank you so much for taking the time to review our rebuttal. We sincerely appreciate your positive feedback and for raising the score.

---

### Official Review · Reviewer_J4bJ · 2025-10-30

**Soundness:** 3
**Presentation:** 2
**Contribution:** 2
**Rating:** 4
**Confidence:** 3

**Summary:**

This paper introduces ICR, which extracts Principal ICL Directions from attention and adaptively injects them into logits via a lightweight router. Experiments show ICR outperforms prior implicit ICL, remains stable on out-of-distribution tasks, and achieves strong efficiency. It offers a new paradigm with few parameters, zero-shot generalization, and cross-task reusability.

**Strengths:**

- This paper introduces the new paradigm of attention routing, shifting implicit ICL from residual injection to low-rank bias at the logits level, demonstrating clear novelty.

- It achieves consistent gains on open-source models such as Llama2, Qwen2.5, and Llama3.1, showing strong generality and reusability.

**Weaknesses:**

1. The evaluation is limited to classification and reasoning tasks, lacking assessment on open-ended QA and long-context reasoning.

2. Experiments are only conducted on 7B/8B models, without validation on larger-scale LLMs.

3. The router relies solely on a fixed MiniLM encoder for query representations, without examining whether alternative encoders could affect routing quality and generalization.

**Questions:**

see weaknesses

---

> ### Author Response · Authors · 2025-11-21
>
> We thank the reviewer for the valuable and insightful comment. We respond to each comment as follows and sincerely hope that our rebuttal properly addresses your concerns. If so, we would deeply appreciate it if you could raise your score (currently 4). If not, please let us know your further concerns, and we will continue actively responding to your comments and improving our submission.
>
> **Weakness1**
> > The evaluation is limited to classification and reasoning tasks, lacking assessment on open-ended QA and long-context reasoning.
>
> **R1:**
> We thank the reviewer for raising this concern. **We would like to highlight a relevant aspect of the current research landscape to contextualize this point**. According to recent works [1–4] on representation engineering, it remains challenging and unsolved for representation steering and implicit ICL methods to perform well in open-ended generation tasks. This difficulty largely arises from how to apply and maintain steering across multiple generation steps, which likely requires task-specific formulations or step-wise steering mechanisms [3], a direction that deserves dedicated future research. Currently, almost all implicit ICL studies restrict evaluation to classification/reasoning/QA families [4–6]. **Our paper’s focus is to establish label-free generalization of ICL internalization and routing within that family, where we already show transfer to disjoint label spaces and different reasoning operations without using any OOD supervision.** We view extending implicit ICL to open-ended generation tasks as an important open direction beyond our present scope.
>
> However, to directly address the reviewer’s suggestion, we add a small pilot comparison of ICR using Llama3.1-8B with zero/few-shot prompting and representative implicit-ICL baselines on **one long-form generation task**, Xsum (https://huggingface.co/datasets/EdinburghNLP/xsum) [7] and **one open-ended mathematical word-problem reasoning task**, Lila (https://huggingface.co/datasets/allenai/lila) [8], reporting rouge-2 and accuracy respectively. We apply steering at every generation step for both the residual-based baselines and our ICR.
>
> The results show that ICR can still maintain a small **positive** effect over zero-shot prompting, whereas residual-based steering baselines tend to destabilize generation and often perform worse than zero-shot. At the same time, both ICR and the baselines still lag behind few-shot prompting by a clear margin. We view these findings as preliminary evidence that ICR can be **safely** applied to generation tasks without degrading performance, while more targeted methods for controlling steering across generation steps remain an important direction for future work.
>
> We hope this addition can make our generalization claims more transparent and convincing.
>
> | Methods \ Tasks (Metric) | Xsum (Rouge-2) | Lila (Acc.) |
> |--------------------------|----------------|-------------|
> | Zero-shot                | 0.0804         | 0.39        |
> | Few-shot (3-shot)        | 0.1663         | 0.54        |
> | TV                       | 0.0765         | 0.26        |
> | I2CL                     | 0.0789         | 0.28        |
> | LIVE                     | 0.0806         | 0.35        |
> | ICR                      | 0.0829         | 0.41        |
>
> References:
>
> [1] Wehner, Jan et al. “Taxonomy, Opportunities, and Challenges of Representation Engineering for Large Language Models.” arXiv:2502.19649 [cs]. Mar. 2025. DOI: 10.48550/arXiv.2502.19649. URL: http://arxiv.org/abs/2502.19649
>
> [2] Itamar Pres, Laura Ruis, Ekdeep Singh Lubana, and David Krueger. Towards reliable evaluation of behavior steering interventions in LLMs. In Neurips 2024 Workshop MINT: Foundation Model Interventions, 2024. URL https://openreview.net/forum?id=7xJcX2gbm9.
>
> [3] Daniel Scalena, Gabriele Sarti, and Malvina Nissim. Multi-property steering of large language models with dynamic activation composition. arXiv preprint arXiv:2406.17563, 2024.
>
> [4] Zhuowei Li, Zihao Xu, Ligong Han, Yunhe Gao, Song Wen, Di Liu, Hao Wang, and Dimitris N Metaxas. Implicit in-context learning. ICLR, 2025.
>
> [5] Eric Todd, Millicent L Li, Arnab Sen Sharma, Aaron Mueller, Byron C Wallace, and David Bau. Function vectors in large language models. ICLR, 2024.
>
> [6] Futing Wang, Jianhao Yan, Yue Zhang, and Tao Lin. Elicit: Llm augmentation via external incontext capability. ICLR, 2025.
>
> [7] Shashi Narayan, Shay B. Cohen, and Mirella Lapata. 2018. Don’t give me the details, just the summary! topic-aware convolutional neural networks for extreme summarization. EMNLP, 2018
>
> [8] Swaroop Mishra, Matthew Finlayson, et al. Lila: A unified benchmark for mathematical reasoning. EMNLP, 2022.

---

> ### Author Response · Authors · 2025-11-21
>
> **Weakness2**
> > Experiments are only conducted on 7B/8B models, without validation on larger-scale LLMs.
>
>
> **R2:**
> We thank the reviewer for raising this concern. Considering computational cost, evaluating on 7B–8B models is standard practice in the implicit ICL literature, and the baselines we compare against also typically report their main results at this scale in their papers. To further assess robustness across scales, we have evaluated our method on three models ranging from **14B to 32B and 70B** parameters. For Qwen3-14B and Qwen3-32B we set the PID rank to r=8, and for Llama3.1-8B we use r=12. ICR is applied to the last one-fourth of the transformer layers.
>
> The results are shown below:
> | Tasks     | AG   | SST2 | TREC | CSQA | PIQA | SST5 | MR   | MRPC | CB   | COPA | CREAK | AI2SciE | Avg   |
> |-----------|-----:|-----:|-----:|-----:|-----:|-----:|-----:|-----:|-----:|-----:|------:|--------:|------:|
> | zero-shot | 69.8 | 80.0 | 55.6 | 78.6 | 83.8 | 44.4 | 72.2 | 75.8 | 85.7 | 94.0 | 76.6  | 91.6    | 75.7  |
> | few-shot  | 77.4 | 85.8 | 79.6 | 82.8 | 85.2 | 48.8 | 88.6 | 79.8 | 87.5 | 96.0 | 62.4  | 98.0    | 81.0  |
> |  FV         |  68.6    |  81.2    |  62.4    |  77.6    | 82.4     |  45.2    |   73.4   |   76.4   |  86.2    |  85.4    |  74.8     |   93.4      |  75.6  |
> |  I2CL        |  69.6    |  83.6    |  64.4    |  79.4    |  84.0    |  46.6    |  73.8    |  76.0    |   87.0   |  87.4    |  77.0     |   95.2      |  77.0 |
> | ICR       | 74.8 | 84.6 | 69.8 | 82.6 | 89.2 | 48.4 | 79.0 | 78.6 | 89.3 | 88.0 | 80.4  | 98.8    | 80.3  |
>
> Table1: Results on **Qwen3 (14B)**.
>
> | Tasks     | AG   | SST2 | TREC | CSQA | PIQA | SST5 | MR   | MRPC | CB   | COPA | CREAK | AI2SciE | Avg |
> |-----------|-----:|-----:|-----:|-----:|-----:|-----:|-----:|-----:|-----:|-----:|------:|--------:|-----:|
> | zero-shot | 69.8 | 83.0 | 51.2 | 76.0 | 78.0 | 46.0 | 95.2 | 75.0 | 91.1 | 96.0 | 51.8  | 85.8    | 74.9 |
> | few-shot  | 84.4 | 89.8 | 77.8 | 86.8 | 89.0 | 43.8 | 99.4 | 76.4 | 91.1 | 98.0 | 51.4  | 80.3    | 80.7 |
> |  FV       |  74.6    |  82.6    |  58.2    |  75.2    |  63.4    |  35.6    |  93.4    |  74.8    |  85.2    |  93.8    |  47.0     |   79.4      |  71.9 |
> |  I2CL     |  78.4    |  85.6    |  64.7    |  74.6    |   74.2   |  38.6    |  94.8    |  76.0    |  89.6    |   94.2   |  52.0     |   84.6      | 75.6 |
> | ICR       | 81.6 | 86.4 | 77.2 | 82.0 | 82.8 | 50.4 | 97.2 | 79.8 | 94.6 | 96.0 | 53.6  | 88.6    | 80.9 |
>
> Table2: Results on **Qwen3 (32B)**.
>
> | Tasks     | AG   | SST2 | TREC | CSQA | PIQA | SST5 | MR   | MRPC | CB   | COPA | CREAK | AI2SciE | Avg  |
> |-----------|-----:|-----:|-----:|-----:|-----:|-----:|-----:|-----:|-----:|-----:|------:|--------:|-----:|
> | zero-shot | 48.8 | 93.2 | 62.6 | 80.2 | 70.4 | 44.0 | 82.0 | 71.4 | 91.0 | 96.0 | 92.6  | 68.6    | 77.6 |
> | few-shot  | 70.8 | 91.0 | 68.0 | 83.2 | 88.6 | 46.4 | 85.2 | 78.6 | 92.9 | 97.0 | 88.0  | 96.2    | 82.2 |
> | FV        |  52.4    |  86.4    |  58.4    |  75.6    | 71.2     |  42.6    | 78.8     | 68.4     |   85.6   |  86.4    |  85.8     |   80.8      | 72.7  |
> | I2CL      |  62.6    |  88.8    | 63.0     |  73.8    |  75.4    | 46.8     |  77.6    | 70.0     |  90.2    |   89.0   | 88.0      |   84.4      | 75.8  |
> | ICR       | 66.4 | 93.8 | 66.0 | 82.4 | 83.2 | 48.4 | 86.8 | 80.2 | 93.4 | 92.0 | 93.2  | 92.0    | 81.5 |
>
> Table3: Results on **Llama3.1 (70B)**.
>
> Across the three models with increasing scale, ICR consistently outperforms the two residual-injection baselines. It approaches few-shot performance in in-domain settings and typically surpasses cross-task few-shot prompting in OOD settings, with only a few exceptions where the large-scale model is already very strong and results become slightly unstable. These trends further validate that ICR achieves stable transfer by avoiding reliance on noisy cross-task ICDs, and demonstrate its effectiveness across models of different scales.
>
> We have added this to Appendix E.3 in revision.

---

> ### Author Response · Authors · 2025-11-21
>
> **Weakness3**
> > The router relies solely on a fixed MiniLM encoder for query representations, without examining whether alternative encoders could affect routing quality and generalization.
>
> **R3:**
> We appreciate the reviewer’s insightful comment regarding the role of the frozen text encoder. To address this concern, we conduct an ablation with Llama2-7B in which we replace all-MiniLM-L6-v2 with a stronger encoder, all-mpnet-base-v2, which has more layers and a higher embedding dimension (768 > 384). This allows us to directly assess whether routing quality and cross-domain generalization depend on the specific choice of encoder.
>
> The results show that the stronger encoder mpnet yields **slightly better performance** in both ID and OOD settings, while the overall trend and relative performance of ICR remain consistent. This indicates that (i) ICR’s generalization behavior is **robust to the encoder choice** rather than tied to a particular model, and  (ii) the router is able to **effectively leverage the semantic representations** provided by the frozen encoder. As expected, larger encoders offer marginally better semantic retrieval at the cost of slower usage, so the choice involves a tradeoff between speed and capacity.
>
> We have included this encoder ablation in Appendix G.5 in the revision.
>
> | encoder\task      | agnews | sst2 | trec | csqa | piqa | sst5 | mr   | mrpc | cb   | copa | creak | ai2scie | Avg  |
> |-----------|-------:|-----:|-----:|-----:|-----:|-----:|-----:|-----:|-----:|-----:|------:|--------:|-----:|
> | all-MiniLM-L6-v2      | 86.6 | 86.4 | 83.8 | 24.8 | 57.0 | 38.6 | 79.8 | 53.4 | 46.4 | 68.0 | 56.4  | 37.2    | 59.9 |
> | all-mpnet-base-v2     | 86.8 | 87.0 | 88.2 | 25.0 | 58.6 | 37.0 | 86.4 | 56.6 | 48.2 | 64.0 | 57.0  | 38.0    | 61.1 |
> | Δ                   | +0.2 | +0.6 | +4.4 | +0.2 | +1.6 | -1.6 | +6.6 | +3.2 | +1.8 | -4.0 | +0.6  | +0.8    | +1.2 |

---

> > ### Comment · Reviewer_J4bJ · 2025-11-25
> >
> > I appreciate the thorough explanations and newly added experiments. These revisions directly resolve my main concerns. Accordingly, I have updated my score from 4 to 6.

---

> > > ### Author Response · Authors · 2025-11-25
> > >
> > > Thank you very much for taking the time to review our rebuttal. We sincerely appreciate your positive feedback and for raising the score.

---

### Official Review · Reviewer_7Mtp · 2025-11-01

**Soundness:** 3
**Presentation:** 3
**Contribution:** 2
**Rating:** 4
**Confidence:** 3

**Summary:**

This paper introduces In-Context Routing (ICR), a novel approach to improve large language models' in-context learning capabilities without using explicit demonstration examples. ICR extracts generalizable patterns from multi-domain in-context learning by identifying Principal ICL Directions (PIDs) through PCA on attention representations. These patterns are applied via a learnable router that modulates attention logits based on input queries, enabling effective zero-shot inference. Unlike existing vector-based implicit ICL methods that inject task-specific vectors into residual streams, ICR operates at the attention mechanism level, providing better generalization. Experiments on 12 datasets show ICR consistently outperforms baselines, particularly excelling on out-of-domain tasks while maintaining computational efficiency comparable to zero-shot inference.

**Strengths:**

1.  ICR operates at the attention logits level rather than post-hoc residual stream injection, which is more aligned with how ICL fundamentally works through attention mechanisms
2. The paper provides rigorous theoretical grounding using the Spiked Covariance Model and Davis-Kahan theorem to explain why PCA on multi-domain ICL bases can extract generalizable patterns.
3. Instead of additive vector interventions, ICR modulates attention through low-rank modifications to query-key interactions.
4. Novel use of PCA to extract reusable structural directions from cross-domain attention representations

**Weaknesses:**

1. OOD Design Issues:
 The division into "near-OOD" and "far-OOD" seems subjective. For example, why is MRPC (paraphrase detection) considered "near" while CB (NLI) is "far"? Both involve sentence-pair understanding. The "OOD" tasks are still mostly classification/QA tasks from standard NLP benchmarks. True OOD would include fundamentally different task types (e.g., structured prediction, generation, mathematical reasoning). The paper trains on 5 diverse datasets (AGNews, SST-2, TREC, CSQA, PIQA) which already cover sentiment, QA, and classification. This makes the "generalization" less impressive since the model has seen similar task types during training.
2. The technical contributions are relatively incremental: The core idea of routing attention through PCA-extracted directions is reasonable, but the execution lacks the technical depth and innovation expected for a top-tier venue. A stronger contribution would involve more sophisticated pattern extraction, adaptive routing mechanisms, or novel theoretical insights about ICL.
3. The quality of PIDs heavily depends on the diversity and quality of initial ICL prompts, but no guidelines are provided for this critical step

**Questions:**

1. The experiments only test on 7B-8B models. How does ICR scale to larger models (70B+) where ICL behavior might be fundamentally different?
2. No analysis of how PID dimensionality (r) should scale with model size or task complexity
3. Computational cost of extracting PIDs grows with the number of domains, but this overhead isn't thoroughly analyzed when compared with few-shot learning which require zero training but may cost at inference.

---

> ### Author Response · Authors · 2025-11-21
>
> We thank the reviewer for the valuable and insightful comment. We respond to each comment as follows and sincerely hope that our rebuttal properly addresses your concerns. If so, we would deeply appreciate it if you could raise your score (currently 4). If not, please let us know your further concerns, and we will continue actively responding to your comments and improving our submission.
>
> **Weakness1 (Part1):**
> > OOD Design Issues: The division into "near-OOD" and "far-OOD" seems subjective. For example, why is MRPC (paraphrase detection) considered "near" while CB (NLI) is "far"? Both involve sentence-pair understanding.
>
> **R1:**
> Thank you for this insightful comment. We agree that OOD taxonomy can feel subjective without explicit criteria. To address this concern, we first articulate a clearer framework for how we distinguish ID, near-OOD, and far-OOD. In the context of ICL generalization, we consider three key axes: **(i) the compatibility of the label ontology with the ID family, (ii) the similarity of the required operation or reasoning structure, and (iii) the degree of semantic and domain shift.**
>
> Under this framework, near-OOD tasks are those that deviate from ID along **at most** one of these axes while preserving the overall ICL structure. SST-5, MR, and MRPC fall into this category. SST-5 extends SST-2 from a binary polarity to a five-point sentiment scale, so the label ontology changes in granularity but remains sentiment-based, while the operation type and domain are essentially identical. MR (Rotten Tomatoes) keeps the same binary sentiment ontology as SST-2 and the same sentence-level classification operation, but shifts the underlying review corpus, so the main change is semantic and domain shift. MRPC, although cast as a sentence-pair paraphrase, still uses a binary decision geometry that is compatible with SST-2 and the A/B decision format of PIQA, and the reasoning required is largely surface-level alignment, such as lexical and syntactic rewrites. Thus, it introduces a mild change in operation type but remains close to ID along the label ontology and general language style.
>
> By contrast, far-OOD tasks shift along **multiple** axes at once. CB, COPA, CREAK, and AI2SciE all introduce new label inventories and reasoning structures that are not present in any ID dataset, together with nontrivial semantic shifts. CB is a three-way NLI task with labels entailment, contradiction, and neutral, which do not align with any ID label space, and it requires **directional inference** from premise to hypothesis. COPA formulates explicit **causal reasoning** over alternatives, which differs from the recognition-style decisions in ID and entails a different type of relational reasoning. CREAK focuses on claim verification, relying on world knowledge and on reasoning about when seemingly **plausible statements fail in specific cases**, rather than on shallow sentence-level judgments. AI2SciE requires **scientific explanatory** reasoning over domain-specific content that is absent from the ID datasets. Together, these shifts in label ontology, operation type, and semantics alter the effective ICL geometry in more than one dimension, which is why we categorize them as far-OOD.
>
> **For MRPC and CB specifically**, MRPC remains close to ID because it is a **symmetric** binary similarity decision over sentence pairs, with a label ontology and decision geometry compatible with SST-2 and the A/B choice format in PIQA, and it mostly relies on **surface-level alignment** like lexical/syntactic rewrites. By contrast, CB introduces a three-way NLI label space (entailment/contradiction/neutral) absent from ID and requires **directional** premise–hypothesis inference with richer pragmatic cues (e.g., negation, quantification), which shifts both the ontology and reasoning operation.
>
> We have included a detailed description of this three-axis framework in Appendix D.3.1 in the revision.

---

> ### Author Response · Authors · 2025-11-21
>
> **Weakness1 (Part2):**
> > The "OOD" tasks are still mostly classification/QA tasks from standard NLP benchmarks. True OOD would include fundamentally different task types (e.g., structured prediction, generation, mathematical reasoning). The paper trains on 5 diverse datasets (AGNews, SST-2, TREC, CSQA, PIQA) which already cover sentiment, QA, and classification. This makes the "generalization" less impressive since the model has seen similar task types during training.
>
> **R2:**
> **Clarification of OOD evaluation**
>
> We would like to clarify the rationale of our OOD design and explain why the chosen OOD tasks constitute a meaningful evaluation setting for ICR.
>
> Although the used OOD tasks are mostly standard NLP benchmarks, the label disjointness, reasoning shift, and semantic mismatch create genuine distribution shift (as illustrated above), which is the setting ICR is designed to handle. **This style of near/far OOD construction is actually common in the NLP OOD literature** [1,2].
>
> We acknowledge that “true OOD” tasks could include fundamentally different paradigms. However, as shown in recent work [3–6], it remains challenging and **unsolved** for representation steering and implicit ICL methods to perform well in long-form generation tasks. This difficulty largely arises from how to apply and maintain steering across multiple generation steps, which likely requires task-specific formulations or step-wise steering mechanisms [5]. Currently, almost all implicit ICL studies restrict evaluation to classification/reasoning/QA families [6–8]. **Our paper’s focus is to establish label-free generalization of ICL internalization and routing within that family, where we already show transfer to disjoint label spaces and different reasoning operations without using any OOD supervision.** We view extending implicit ICL to long generation and structured outputs as an important open direction beyond our present scope.
>
> Nevertheless, for a more rigorous evaluation, we add a pilot test of ICR using Llama3.1-8B with zero/few-shot prompting and representative implicit-ICL baselines on **one long-form generation task**, Xsum (https://huggingface.co/datasets/EdinburghNLP/xsum) [9] and **one open-ended mathematical reasoning task**, Lila (https://huggingface.co/datasets/allenai/lila) [10], reporting rouge-2 and accuracy, respectively. We do not include a structured prediction task, as such tasks typically require task-specific syntactic or structural cues (either via demonstrations or carefully engineered prompts), which fall far outside the scope of our current work. We apply steering at every generation step for the residual-based baselines and our ICR.
>
> The results show that ICR can still maintain a small **positive** effect over zero-shot prompting, whereas residual-based steering baselines tend to destabilize generation and often perform worse than zero-shot. At the same time, both ICR and the baselines still lag behind few-shot prompting by a clear margin. We view these findings as preliminary evidence that ICR can be **safely** applied to generation tasks without degrading performance, while more targeted methods for controlling steering across generation steps remain an important future work.
>
> We hope this addition can make our generalization claims more transparent and convincing.
>
> | Methods \ Tasks (Metric) | Xsum (Rouge-2) | Lila (Acc.) |
> |--------------------------|----------------|-------------|
> | Zero-shot                | 0.0804         | 0.39        |
> | Few-shot (3-shot)        | 0.1663         | 0.54        |
> | TV                       | 0.0765         | 0.26        |
> | I2CL                     | 0.0789         | 0.28        |
> | LIVE                     | 0.0806         | 0.35        |
> | ICR                      | 0.0829         | 0.41        |

---

> ### Author Response · Authors · 2025-11-21
>
> References:
>
> [1] Jiacheng Liu, Wenya Wang, Dianzhuo Wang, Noah A. Smith, Yejin Choi, and Hannaneh Hajishirzi. Vera: A general-purpose plausibility estimation model for commonsense statements. EMNLP, 2023.
>
> [2] Dan Hendrycks, Mantas Mazeika, and Thomas Dietterich. Deep anomaly detection with outlier exposure. ICLR, 2019.
>
> [3] Wehner, Jan et al. Taxonomy, Opportunities, and Challenges of Representation Engineering for Large Language Models. arXiv:2502.19649 [cs]. Mar. 2025. DOI: 10.48550/arXiv.2502.19649. URL: http://arxiv.org/abs/2502.19649
>
> [4] Itamar Pres, Laura Ruis, Ekdeep Singh Lubana, and David Krueger. Towards reliable evaluation of behavior steering interventions in LLMs. In Neurips 2024 Workshop MINT: Foundation Model Interventions, 2024.
>
> [5] Daniel Scalena, Gabriele Sarti, and Malvina Nissim. Multi-property steering of large language models with dynamic activation composition. arXiv preprint arXiv:2406.17563, 2024.
>
> [6] Zhuowei Li, Zihao Xu, Ligong Han, Yunhe Gao, Song Wen, Di Liu, Hao Wang, and Dimitris N Metaxas. Implicit in-context learning. ICLR, 2025.
>
> [7] Eric Todd, Millicent L Li, Arnab Sen Sharma, Aaron Mueller, Byron C Wallace, and David Bau. Function vectors in large language models. ICLR, 2024.
>
> [8] Futing Wang, Jianhao Yan, Yue Zhang, and Tao Lin. Elicit: Llm augmentation via external incontext capability. ICLR, 2025.
>
> [9] Shashi Narayan, Shay B. Cohen, and Mirella Lapata. 2018. Don’t give me the details, just the summary! topic-aware convolutional neural networks for extreme summarization. EMNLP, 2018.
>
> [10] Swaroop Mishra, Matthew Finlayson, et al. Lila: A unified benchmark for mathematical reasoning. EMNLP, 2022.
>
> **Weakness2**
> > The technical contributions are relatively incremental: The core idea of routing attention through PCA-extracted directions is reasonable, but the execution lacks the technical depth and innovation expected for a top-tier venue. A stronger contribution would involve more sophisticated pattern extraction, adaptive routing mechanisms, or novel theoretical insights about ICL.
>
> **R3:**
> We appreciate the reviewer’s concern. We would like to clarify first that the execution-level choices (e.g., extraction of directions with PCA) are not intended to be the core contribution of ICR. **The technical depth of our work lies not in the implementations like PCA  but in uncovering how attention information and structure can be captured and manipulated inside LLMs, and in demonstrating that the structure can be harnessed to achieve train-once, label-free generalization across tasks.** Existing pipelines for implicit ICL all rely on residual-stream editing and task-specific calibration, none of which provide a mechanism for transferring ICL patterns across domains without labels. We also note that other reviewers characterized the method as “revolutionary” (reviewer LNqB) and as “demonstrating clear novelty” (reviewer J4bJ), which aligns with our view that ICR goes beyond an incremental modification.
>
> Our key innovation, which constitutes a conceptual advance, is identifying the attention-logits space as the effective locus for internalizing ICL patterns. **This is a novel departure from prior implicit ICL works that attempt to operate on residual flows.** We show that logits-space routing exposes reusable structural directions (PIDs), and we introduce a query-conditioned router that adaptively composes these directions with head-level signals to steer attention in a low-rank manner. While pattern extraction is not sophisticated, the novelty is in discovering **what to extract, where to inject it, and how to condition its use on each test-time query.** This enables the generalization behavior that prior work could not achieve.
>
> Empirically, this design consistently outperforms residual-based implicit ICL baselines, **especially under cross-domain and far-OOD shifts where existing methods are inherently not suitable to handle**, which validates that our mechanism is capturing a genuine structural property of ICL rather than yielding incremental improvements. Taking together the train-once, label-free objective, the logits-space locus, and the query-adaptive composition, ICR offers a novel and practically meaningful way of leveraging ICL beyond the labeled ID regime, rather than an incremental modification.

---

> ### Author Response · Authors · 2025-11-21
>
> **Weakness3**
> > The quality of PIDs heavily depends on the diversity and quality of initial ICL prompts, but no guidelines are provided for this critical step.
>
> **R4:**
> We thank the reviewer for raising this point. **We do provide guidance for constructing ICL prompts used in PID extraction.** In Section 4.1 (paragraph “Cross-domain collections”), we state: “*For each dataset, we construct ICL prompts by first sampling a query and a balanced set of ICDs, both from the training split, where 5 ICDs are drawn from each class of the same dataset. We construct 10k prompts for AGNews and 5k prompts for each of the remaining datasets.*” We also direct readers at the start of Section 4.3 to Appendix G.2, where we **ablate ICDs sampling strategies and few-shot counts**. In Appendix D.1, we explain the **proportion of ICL prompts from different domains**. Additionally, we provide the complete **prompt templates** used for PIDs extraction (and inference) in Appendix D.3.1 Table 6, which ensures consistent and well-formatted prompt quality.
>
> Appendix G.2 compares our balanced sampling with a similarity-based strategy that selects ICDs semantically close to the query via embedding similarity. With the same total number of ICDs, similarity sampling can approach our method on ID and near-OOD, but it underperforms on far-OOD. This indicates that, while effective in vanilla ICL, similarity-based selection tends to overfit query-local patterns rather than capture cross-domain invariances, and is therefore less suitable for our generalizable pipeline. We also vary the shot count and find that 5 per class is a good trade-off. These results, summarized in Table 10, support our recommendation to emphasize **balance and diversity** when constructing prompts for PID extraction.
>
> Appendix D.1 clarifies that the five diverse in-domain datasets for PIDs extraction fall into three broad categories: topic-level (AGNews), sentence-semantics level (SST-2 and TREC), and reasoning-oriented QA (CSQA and PIQA). To capture general ICL patterns rather than bias toward a particular sub-pattern, we sample prompts in a balanced manner across these categories.
>
> We placed those details in the appendix due to page limits. We fully agree that these are important guidelines, and we have made the reference to the corresponding appendix more explicit in Section 4.1.

---

> ### Author Response · Authors · 2025-11-21
>
> **Question1**
> > The experiments only test on 7B-8B models. How does ICR scale to larger models (70B+) where ICL behavior might be fundamentally different?
>
> **R5:**
> We thank the reviewer for raising this concern. Considering computational cost, evaluating on 7B–8B models is standard practice in the implicit ICL literature, and the baselines we compare against also typically report their main results at this scale in their papers. To further assess robustness across scales, we have evaluated our method on three models ranging from **14B to 32B and 70B** parameters. For Qwen3-14B and Qwen3-32B we set the PID rank to r=8, and for Llama3.1-70B we use r=12. ICR is applied to the last one-fourth of the transformer layers.
>
> The results are shown below:
> | Tasks     | AG   | SST2 | TREC | CSQA | PIQA | SST5 | MR   | MRPC | CB   | COPA | CREAK | AI2SciE | Avg   |
> |-----------|-----:|-----:|-----:|-----:|-----:|-----:|-----:|-----:|-----:|-----:|------:|--------:|------:|
> | zero-shot | 69.8 | 80.0 | 55.6 | 78.6 | 83.8 | 44.4 | 72.2 | 75.8 | 85.7 | 94.0 | 76.6  | 91.6    | 75.7  |
> | few-shot  | 77.4 | 85.8 | 79.6 | 82.8 | 85.2 | 48.8 | 88.6 | 79.8 | 87.5 | 96.0 | 62.4  | 98.0    | 81.0  |
> |  FV         |  68.6    |  81.2    |  62.4    |  77.6    | 82.4     |  45.2    |   73.4   |   76.4   |  86.2    |  85.4    |  74.8     |   93.4      |  75.6  |
> |  I2CL        |  69.6    |  83.6    |  64.4    |  79.4    |  84.0    |  46.6    |  73.8    |  76.0    |   87.0   |  87.4    |  77.0     |   95.2      |  77.0 |
> | ICR       | 74.8 | 84.6 | 69.8 | 82.6 | 89.2 | 48.4 | 79.0 | 78.6 | 89.3 | 88.0 | 80.4  | 98.8    | 80.3  |
>
> Table1: Results on **Qwen3 (14B)**.
>
> | Tasks     | AG   | SST2 | TREC | CSQA | PIQA | SST5 | MR   | MRPC | CB   | COPA | CREAK | AI2SciE | Avg |
> |-----------|-----:|-----:|-----:|-----:|-----:|-----:|-----:|-----:|-----:|-----:|------:|--------:|-----:|
> | zero-shot | 69.8 | 83.0 | 51.2 | 76.0 | 78.0 | 46.0 | 95.2 | 75.0 | 91.1 | 96.0 | 51.8  | 85.8    | 74.9 |
> | few-shot  | 84.4 | 89.8 | 77.8 | 86.8 | 89.0 | 43.8 | 99.4 | 76.4 | 91.1 | 98.0 | 51.4  | 80.3    | 80.7 |
> |  FV       |  74.6    |  82.6    |  58.2    |  75.2    |  63.4    |  35.6    |  93.4    |  74.8    |  85.2    |  93.8    |  47.0     |   79.4      |  71.9 |
> |  I2CL     |  78.4    |  85.6    |  64.7    |  74.6    |   74.2   |  38.6    |  94.8    |  76.0    |  89.6    |   94.2   |  52.0     |   84.6      | 75.6 |
> | ICR       | 81.6 | 86.4 | 77.2 | 82.0 | 82.8 | 50.4 | 97.2 | 79.8 | 94.6 | 96.0 | 53.6  | 88.6    | 80.9 |
>
> Table2: Results on **Qwen3 (32B)**.
>
> | Tasks     | AG   | SST2 | TREC | CSQA | PIQA | SST5 | MR   | MRPC | CB   | COPA | CREAK | AI2SciE | Avg  |
> |-----------|-----:|-----:|-----:|-----:|-----:|-----:|-----:|-----:|-----:|-----:|------:|--------:|-----:|
> | zero-shot | 48.8 | 93.2 | 62.6 | 80.2 | 70.4 | 44.0 | 82.0 | 71.4 | 91.1 | 96.0 | 92.6  | 68.6    | 75.1 |
> | few-shot  | 70.8 | 91.0 | 68.0 | 83.2 | 88.6 | 46.4 | 85.2 | 78.6 | 92.9 | 97.0 | 88.0  | 96.2    | 82.2 |
> | FV        |  52.4    |  86.4    |  58.4    |  75.6    | 71.2     |  42.6    | 78.8     | 68.4     |   85.6   |  86.4    |  85.8     |   80.8      | 72.7  |
> | I2CL      |  62.6    |  88.8    | 63.0     |  73.8    |  75.4    | 46.8     |  77.6    | 70.0     |  90.2    |   89.0   | 88.0      |   84.4      | 75.8  |
> | ICR       | 66.4 | 93.8 | 66.0 | 82.4 | 83.2 | 48.4 | 86.8 | 80.2 | 93.4 | 92.0 | 93.2  | 92.0    | 81.5 |
>
> Table3: Results on **Llama3.1 (70B)**.
>
> Across the three models with increasing scale, ICR consistently outperforms the two residual-injection baselines. It approaches few-shot performance in in-domain settings and typically surpasses cross-task few-shot prompting in OOD settings, with only a few exceptions where the large-scale model is already very strong and results become slightly unstable. These trends further validate that ICR achieves stable transfer by avoiding reliance on noisy cross-task ICDs, and demonstrate its effectiveness across models of different scales.
>
> We have added this to Appendix E.3 in revision.

---

> ### Author Response · Authors · 2025-11-21
>
> **Question2**
> > No analysis of how PID dimensionality (r) should scale with model size or task complexity
>
> **R6:**
> We thank the reviewer for pointing out this analysis. We have conducted two sets of additional ablations to study how PID dimensionality r interacts with model size and task complexity. The average accuracy over all 12 datasets is reported.
>
> Model Size:
>
> We evaluate ICR on three backbone models with increasing scale: Llama2-7B, Qwen3-14B, and Qwen3-32B. For each backbone, we vary the PID dimensionality r∈{4,8,12}. On Llama2-7B and Qwen3-14B, r=8 gives the best overall performance, with both r=4 and r=12 worse. When moving to Qwen3-32B, the best setting shifts to r=12, with r=12>r=8>r=4. This simple scaling experiment suggests a preliminary trend that **the optimal rank r tends to increase as the backbone model becomes larger**.
>
> | r \ model | LLaMA2-7B | Qwen3-14B | Qwen3-32B |
> |-----------|----------:|----------:|----------:|
> | 4         | 57.8      | 76.0      | 78.2      |
> | 8         | **59.9**  | **80.3**  | 80.9      |
> | 12        | 50.2      | 72.2      | **82.5**  |
>
> Task Complexity:
>
> To probe the effect of task complexity, we run an additional ablation on Llama2-7B under two PID-collection regimes: (i) the default setting using all five ID datasets, and (ii) a simplified setting where PIDs are extracted only from three ID tasks (AGNews, SST-2, TREC). Again, we vary r∈{4,8,12}. In both regimes, r=8 remains the best choice overall. However, with the simpler PID-collection set, the performance gap between r=4 and r=8 becomes smaller, while r=12 underperforms in both regimes. This pattern tentatively indicates that, compared to task complexity, **model size exerts a stronger influence on the optimal rank**. Instead, for relatively simple task combinations, a smaller r (e.g., 4) can approximate the performance of r=8 more **closely**.
>
> | task \ r                      | 4    | 8    | 12   |
> |------------------------------|-----:|-----:|-----:|
> | agnews-sst2-trec             | 56.3 | **57.0** | 48.6 |
> | agnews-sst2-trec-csqa-piqa   | 57.8 | **59.9** | 50.2 |
>
> These ablations provide initial evidence on how r should be chosen with respect to model scale and task mixture. A more systematic and quantitative characterization of the relationship between r, model size, and task complexity is an interesting direction for future work. In this paper, we treat r as a hyperparameter selected via validation rather than claiming a fully principled scaling law.
>
> **Question3**
> > Computational cost of extracting PIDs grows with the number of domains, but this overhead isn't thoroughly analyzed when compared with few-shot learning which require zero training but may cost at inference.
>
> **R7:**
> Thank you for raising this point. We want to clarify that while the computational overhead of PID extraction grows with the number of in-domain datasets, this cost is **one-time and amortized**. The dominant cost of PIDs extraction comes from collecting Q/K representations via a few-shot forward pass, and the PCA step itself is negligible relative to model inference. **Directly comparing this one-time PID extraction cost to the per-instance few-shot ICL inference cost can be misleading, since the latter is incurred for every test example**.
>
> Few-shot ICL requires repeatedly evaluating long prompts with demonstrations, and this overhead scales with the number of downstream tasks and queries. In effect, the total cost of few-shot ICL grows roughly linearly with the volume of test-time usage, whereas ICR pays a single upfront cost and then runs inference in a zero-shot manner. Moreover, as we discuss in Section 2.3, increasing the number of in-domain datasets properly helps us extract PIDs that are more stable and more generalizable, resulting in a router that benefits a broader range of downstream tasks. In other words, the one-time few-shot cost for PIDs extraction, even as we add more in-domain domains, allows the system to avoid repeated few-shot inference costs for a larger number of future tasks.
>
> Thus, the computational cost of extracting PIDs as the number of domains grows is a one-time few-shot investment that is amortized over future tasks and enables broader reuse at inference, rather than a recurring overhead as in few-shot prompting.

---

> ### Author Response · Authors · 2025-11-27
>
> Dear Reviewer 7Mtp,
>
> We kindly follow up to see if you had a chance to review our rebuttal. We would greatly appreciate any additional feedback you may have. If our clarifications address your concerns, we would be grateful if you could consider updating your score.
>
> Thank you again for your time and consideration!
>
> Best regards,
>
> Authors

---

### Official Review · Reviewer_sqZY · 2025-11-02

**Soundness:** 2
**Presentation:** 3
**Contribution:** 2
**Rating:** 6
**Confidence:** 3

**Summary:**

This paper proposes In Context Routing (ICR) for implicit in context learning. Instead of adding demonstration tokens to the prompt or adding shift vectors in the residual stream, the method extracts Principal ICL Directions from multi domain explicit ICL runs by doing PCA at each layer. A small router then maps a new input to layer wise weights and per head gates. During inference the method adds a low rank input conditioned bias to the attention logits. The authors suggest a kernel view interpretation plus low rank reparameterization. Experiments on diverse datasets with several open models show consistent gains over other implicit ICL and few shot ICL and stronger OOD robustness.

**Strengths:**

- Practical efficiency: No prompt length increase and no weight updates in the base model (compared to vanilla ICL). The added compute is low rank and local to the attention logits, which is friendly to deployment compared with long demonstrations or broad fine tuning.
- Clear design shift in implicit ICL: The key novelty is the move from post hoc residual steering to structural routing at the attention logits. This places the intervention exactly where ICL mechanisms operate and turns implicit ICL into a problem of routing attention paths. This is a fresh axis that is different from LoRA that edits weights and from activation steering that edits residuals.
- Empirical breadth and stability: The method wins against several implicit ICL baselines on both in domain and out of domain sets and shows fewer collapses below zero shot. It sometimes matches or beats few shot prompting while keeping zero shot latency and memory.

**Weaknesses:**

- Data and supervision needs: Router training uses labeled data from several domains. The limits of generalization to tasks with new label spaces or to settings without labels are not fully explored. It remains unclear how far the train once and reuse promise extends.
- Information usage in PID extraction: Using only the last token Q and K may underuse the rich structure inside demonstrations. The paper argues it is sufficient as an integration point, but alternative choices like pooling across several tokens or using attention rollouts could strengthen the claim.

**Questions:**

- Why restrict PID extraction to the last token only. Have you tested using several recent tokens or a learned pooling over the demonstration region, and how would that affect out of domain robustness and interpretability?

---

> ### Author Response · Authors · 2025-11-21
>
> We thank the reviewer for the valuable and insightful comments. We respond to each comment as follows and sincerely hope that our rebuttal properly addresses your concerns. If so, we would deeply appreciate it if you could raise your score. If not, please let us know your further concerns, and we will continue actively responding to your comments and improving our submission.
>
> **Weakness1:**
> > Data and supervision needs: Router training uses labeled data from several domains. The limits of generalization to tasks with new label spaces or to settings without labels are not fully explored. It remains unclear how far the train once and reuse promise extends.
>
> **R1:**
> We appreciate the concern and apologize for any confusion. The limitation the reviewer highlights regarding generalization to settings with new or missing label spaces actually characterizes much prior work, and addressing it is a core goal of ICR. Our method is explicitly designed to generalize to tasks whose label spaces are new or unavailable at test time.
>
> To clarify, ICR uses supervision once on in-distribution (ID) datasets to (i) extract PIDs and (ii) train the router. No labels from any OOD dataset (near/far) are used at any stage. At inference time, the router is label-agnostic. It conditions low-rank routing solely on the query’s representation geometry within the PID subspace, independent of the target task’s label inventory. Consequently, the train-once components (PIDs + router) can be reused on tasks with new or unknown label spaces.
>
> Empirically, our OOD evaluations already instantiate this setting where OOD tasks have disjoint label spaces and no OOD labels are used for tuning (Tables 1–2). Thus, provided labeled ID data to set up PID extraction and router training, the resulting ICR router that is frozen at test time generalizes and applies to tasks with new label spaces or even without labels. We have made this point more explicit in the revision (Section 3.4).

---

> ### Author Response · Authors · 2025-11-21
>
> **Weakness2 & Question1:**
> > Information usage in PID extraction: Using only the last token Q and K may underuse the rich structure inside demonstrations. The paper argues it is sufficient as an integration point, but alternative choices like pooling across several tokens or using attention rollouts could strengthen the claim.
>
> > Why restrict PID extraction to the last token only. Have you tested using several recent tokens or a learned pooling over the demonstration region, and how would that affect out of domain robustness and interpretability?
>
> **R2:**
> Thank you for raising this important point. We agree that exploring alternative ways of extracting Q/K representations during PID extraction is meaningful, and we have added three additional variants to examine this. Specifically, in addition to the last-token extraction used in the main paper, we now test (i) mean pooling over the last 4 tokens, (ii) mean pooling over the last 8 tokens, and (iii) an attention-rollout–based pooling that aggregates token-level Q/K using attention-flow weights computed across all layers. Conceptually, the rollout variant constructs a cumulative attention map by multiplying layer-wise attention matrices and uses the resulting contribution scores to weight each token’s Q/K before pooling.
>
> Regarding learned pooling, we believe it is less practical in our setting. Because PID extraction is implemented as an offline PCA-based stage on top of pre-aggregated Q/K representations, the loss used to train the ICR router does not backpropagate through this step. It may require defining a new loss function in the PIDs extraction stage. This would complicate the pipeline and deviate from our goal of learning only a frozen and reusable router on top of fixed PIDs. We therefore focus our analysis on nonparametric pooling strategies that preserve the simplicity of ICR.
> Experimental results with Llama2-7B for the above variants are reported below.
> | Method           | AG   | SST2 | TREC | CSQA | PIQA | SST5 | MR   | MRPC | CB   | COPA | CREAK | AI2SciE | Avg  |
> |------------------|-----:|-----:|-----:|-----:|-----:|-----:|-----:|-----:|-----:|-----:|------:|--------:|-----:|
> | Last token       | 86.6 | 86.4 | 83.8 | 24.8 | 57.0 | 38.6 | 79.8 | 53.4 | 46.4 | 68.0 | 56.4  | 37.2    | **59.9** |
> | Last 4 token     | 84.6 | 83.2 | 83.8 | 25.6 | 55.0 | 35.6 | 73.8 | 46.6 | 39.3 | 61.0 | 53.4  | 31.2    | 56.1 |
> | Last 8 token     | 67.8 | 77.2 | 71.4 | 23.8 | 54.0 | 27.6 | 70.4 | 45.6 | 28.6 | 62.0 | 55.8  | 31.6    | 51.3 |
> | Attention Rollout| 67.2 | 78.8 | 67.0 | 22.2 | 52.6 | 26.8 | 72.8 | 44.4 | 32.2 | 62.0 | 52.4  | 32.8    | 50.9 |
>
> We also conducted an additional experiment on Qwen3-14B to compare last-token pooling with last-4-token pooling for quick verification.
> | Method        | AG   | SST2 | TREC | CSQA | PIQA | SST5 | MR   | MRPC | CB   | COPA | CREAK | AI2SciE | Avg  |
> |---------------|-----:|-----:|-----:|-----:|-----:|-----:|-----:|-----:|-----:|-----:|------:|--------:|-----:|
> | Last token    | 74.8 | 84.6 | 69.8 | 82.6 | 89.2 | 48.4 | 79.0 | 78.6 | 89.3 | 88.0 | 80.4  | 98.8    | **80.3** |
> | Last 4 token  | 70.6 | 82.8 | 66.8 | 82.0 | 87.4 | 46.6 | 74.2 | 75.6 | 91.1 | 84.0 | 75.6  | 92.0    | 77.4 |
>
>
> Across all benchmarks including ID, near-OOD, and far-OOD, none of the alternative pooling strategies outperform the last-token extraction. Performance **degrades as the pooling window expands**, and the attention-rollout variant yields the weakest results. This pattern suggests that incorporating a broader set of tokens introduces noise from heterogeneous token roles, diluting the ICL-related signal that PIDs aim to isolate. A clear trend emerges that **the more tokens included in the pooling region, the more the essential alignment signal is blurred.**
>
> The effectiveness of the last-token extraction is actually consistent with the functional role of this position in ICL. The final token before answering is where the model synthesizes the full prefix (query and demonstrations) into a single attention computation immediately before prediction. This makes it a coherent integration point where the demonstration-induced structure is concentrated. Moreover, it is precisely the position at which ICR injects its attention-logits bias during inference. **Extracting PIDs from the same locus where the intervention is later applied provides a natural alignment between the raw attention geometry and the added low-rank bias.**
>
> These results indicate that the last-token Q/K captures the most stable and transferable ICL-related structure, while broader pooling mixes in context that is not directly relevant for the ICL computation. From an interpretability perspective, the fact that PIDs extracted at the same position where we intervene work best shows that **ICR is indeed leveraging the attention structure that few-shot ICL forms at this locus.**
>
> We have added this additional experiment to Appendix G.4.

---

### Author Response · Authors · 2025-12-02
**Rebuttal Summary**

Dear AC and reviewers:

Thank you for your careful evaluation of our work. We appreciate the positive comments noted by all four reviewers. They recognized the strength of our ICR method in its design, theoretical grounding, motivation, and novelty, offering remarks such as “**clear design shift**” (sqZY), “**provides rigorous theoretical grounding**” (7Mtp), “**demonstrating clear novelty**” (J4bJ), and even “**revolutionary**” (LNqB). During the rebuttal period, we addressed each reviewer’s concerns in detail and provided new experimental results together with additional analysis. **Two reviewers (J4bJ and LNqB)** have expressed positive feedback toward our rebuttal and **increased their scores (6644 -> 8664)**, on 25 Nov and 24 Nov respectively.

Below, we summarize our rebuttal, beginning with two additional experiments requested by Reviewers 7Mtp and J4bJ.

* **Experiments on open-ended tasks**:
We clarify that extending representation-steering-based implicit ICL to open-ended generation remains an open challenge highlighted in recent related works, as it requires carefully designed multi-step steering. Our study primarily targets classification and reasoning tasks, where implicit ICL methods are typically evaluated, including OOD settings within this scope. Nonetheless, we added experiments on two open-ended generation benchmarks (XSum and Lila). The results show that ICR yields a small improvement over zero-shot prompting, whereas residual-based baselines often destabilize generation. These findings suggest that ICR can be safely applied to generation tasks, while developing principled multi-step steering is an important future direction beyond our present scope.
* **Experiments with larger-scale LLMs**:
We expanded our evaluation to larger models, including Qwen3-14B, Qwen3-32B, and Llama3.1-70B. Across these scales, ICR consistently performs well in most settings, indicating that its benefits are not tied to small models.

More specifically,
| Reviewer| Key Concern| Our response|
|-|-|-|
| sqZY| Data and supervision: How our train-once-and-reuse framework extends?| We clarified in detail that no labels from OOD tasks are used when extracting PIDs or when training the router in our ICR pipeline. Once extracted and trained, ICR can be directly applied to unseen OOD scenarios with entirely new or unavailable label spaces. |
|| PID extraction: Test more choices of the information-usage token position.| We added an ablation study, showing that the final token is the most effective integration point, and that expanding the pooling window generally degrades performance.|
| 7Mtp| OOD Design Issues: A clearer standard for the near/far OOD split is needed.| We articulated our three-axis framework for defining near vs. far OOD and provided detailed justification for the categorization of each dataset.|
| | The execution is technically incremental.|We argued that the technical depth of our contribution does not lie in execution such as PCA extraction, but rather in uncovering how attention information and structure can be captured and manipulated within LLMs, and in showing that this structure enables broad generalization. This represents a clear departure from prior implicit ICL methods that operate on residual flows. Our empirical results further support this distinction, showing stronger performance, particularly in OOD settings where residual-based baselines are inherently limited.|
|| Guidelines for the steps of constructing the initial ICL prompts are needed.| We pointed out that we included detailed descriptions and analysis of our ICL prompt construction in both the main text and the appendix, while we have made the cross-reference to the appendix more explicit in the revision.|
| J4bJ (4->6)| An ablation of the frozen text encoders used in the ICR pipeline is needed.| We replaced the default MiniLM-v2 encoder with a stronger MPNet-v2 encoder and found that ICR’s generalization remains robust, with the stronger encoder yielding slightly better results.|
| LNqB (6->8)| Whether the training required by ICR undermines its advantages over the baselines?| We argued that training-in-the-loop is common in implicit ICL methods, including several baselines, and that our trainable component is only a lightweight MLP-based router. More importantly, ICR follows a cross-task train-once-and-reuse paradigm without any task-specific extraction or training, which ensures a fair comparison with the baselines.|
|| Whether the semantics provided by the text encoder are actually utilized by ICR?| We followed the suggestion to replace the text encoder with a random encoder, and the resulting performance drop confirms that ICR indeed benefits from the semantic structure provided by the encoder.|
|| Writing issues.| We provided detailed clarifications in the revision to address all writing issues.|

Overall, we thank the AC and reviewers for their time and effort in helping improve this work.

Best,

Authors

---

### Meta-Review · Area_Chair_e7VW · 2025-12-23

**Summary:**

The paper proposes In-Context Routing (ICR), a method for implicit In-Context Learning (ICL) that extracts "Principal ICL Directions" (PIDs) from attention to modulate these logits during inference . The authors frame this as a "train-once-and-reuse" approach designed to generalize better to Out-Of-Domain (OOD) tasks than prior vector-based methods.

While reviewers LNqB, J4bJ, and sqZY were learning positively, the AC’s recommendation is Reject. This decision is informed by the "incremental" nature of the contribution (Reviewer 7Mtp sharing this concern), and is reinforced by a critical examination of the results. Specifically, on stronger base models (e.g., Llama 3.1-8B in Table 7), the method frequently underperforms Few-Shot (FS) prompting and oftentimes stays on par with zero-shot (ZS). Moreover, the proposed method requires an involved training procedure. These concerns limit the potential impact of the paper, even though the proposed design could moderately save some compute during inference.

**Reviewer Concerns:**

Concerns Addressed by Rebuttal:
- Model Scale: Reviewers 7Mtp and J4bJ  questioned the limitation to 7B/8B models. The authors successfully provided results on Qwen3-14B, Qwen3-32B, and Llama3.1-70B, satisfying this specific request.
- Task Diversity: Reviewer J4bJ noted the lack of open-ended generation tasks. The authors added experiments on XSum and Lila , which the reviewer accepted as resolving the concern.
- Mechanistic Design: Reviewer sqZY questioned the PID extraction method , and LNqB questioned the necessity of the text encoder. The authors provided ablations (e.g., "last-token" extraction vs. pooling , random vs. trained encoder ) that effectively justified these design choices to the reviewers' satisfaction.

Outstanding Concerns:
- Marginal Utility and Heavier Training: Reviewer 7Mtp characterized the technical contribution as "relatively incremental". The method shows negative transfer, and failure to match few-shot prompting. Reviewer LNqB also raised concerns about comparing computationally heavier methods (ICR) to lighter baselines. Given that ICR requires an offline training phase and storage of PIDs, its failure to consistently match the few-shot prompting baseline undermines its practical value.

**Reviewer Scores:**

Reviewer LNqB (6 to 8): This reviewer initially had the most positive rating and had since raised their score to 8 in the rebuttal phase.

Reviewer J4bJ (4 to 6): This reviewer raised their score based on the inclusion of generation tasks.

Reviewer 7Mtp (4 to 4): The rebuttal is likely not going to convince this reviewer to upgrade their score (OOD Design Issues, incremental contribution).

Reviewer sqZY (6 to 6): The rebuttal is relevant to the reviewer’s comment but not likely to swing their opinion large enough for a score upgrade.

---

### Decision · Program_Chairs · 2026-01-26

Reject